# Meta-Black-Box Optimization Can Do Search Guidance for Expensive Constrained Multi-Objective Optimization

Yukun Du[1]   Haiyue Yu[1]   Jiang Jiang[1]   Shuaiwen Tang[1]
Xiaotong Xie[1]   Haobo Liu[1]   Chongshuang Hu[1]   Shengkun Chang[1]

## Abstract

Existing Meta-Black-Box Optimization (MetaBBO) methods focus on *how to search* when controlling optimizers, but largely overlook *where to search*. We propose MetaSG-SAEA, a bi-level MetaBBO framework for expensive constrained multi-objective optimization problems (ECMOPs), in which a meta-policy provides search guidance to the low-level Surrogate-Assisted Evolutionary Algorithm (SAEA). To achieve this, we introduce Max–Min Constraint-Calibrated Inequality (MM-CCI), a compact, problem-agnostic region abstraction that maps heterogeneous constraint evaluations to an ordered scalar level; we further provide a theoretical analysis of its fundamental properties. Building on this region abstraction, we adopt diffusion-based population initialization to translate the meta-policy's region-level guidance into solution-level priors for the SAEA. To make MetaSG-SAEA scalable, we construct an attention-based state representation across varying problem dimensions, population sizes, and numbers of objectives and constraints. Experimental results demonstrate that MetaSG-SAEA outperforms state-of-the-art baselines across diverse benchmarks and exhibits the ability to generalize across problem distributions.

## 1. Introduction

Black-Box Optimization (BBO) considers problems where the objective is accessible only through function evaluations, without analytical forms or gradients. This setting has motivated extensive work in Evolutionary Computation

[1]National University of Defense Technology, Changsha, Hunan, China. Correspondence to: Haiyue Yu <yuhaiyue09@nudt.edu.cn>.

*Proceedings of the 43rd International Conference on Machine Learning*, Seoul, South Korea. PMLR 306, 2026. Copyright 2026 by the author(s).

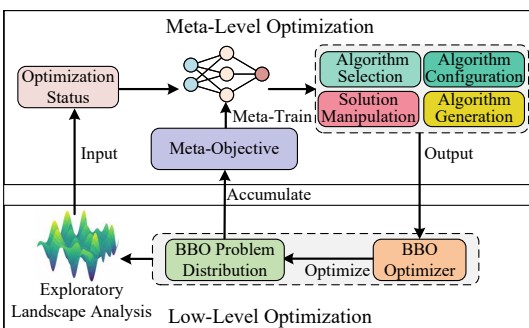

*Figure 1.* Bi-level paradigm of existing MetaBBOs.

(EC), leading to many effective, human-designed black-box optimizers (Slowik & Kwasnicka, 2020; Zhan et al., 2022). However, their transferability is often limited, as an optimizer that performs well on one class of problems may degrade on others and typically requires substantial manual adaptation (Eiben & Smit, 2011; Guo et al., 2024b; Zhan et al., 2025).

To improve transferability, recent Meta-Black-Box Optimization (MetaBBO) methods commonly follow a bi-level paradigm (see Figure 1), where a neural policy is meta-trained over a task distribution to output control decisions for a low-level BBO optimizer (Ma et al., 2023; Zheng et al., 2026). The policy typically takes Exploratory Landscape Analysis (ELA) descriptors as inputs to summarize the current search state and inform its decisions (Mersmann et al., 2011; Ma et al., 2025a). Based on their control mechanisms and roughly ordered by increasing control-space complexity (Yang et al., 2025; Ma et al., 2025c), representative approaches include Algorithm Selection (AS), Algorithm Configuration (AC), Solution Manipulation (SM), and Algorithm Generation (AG). Despite this progress, most existing MetaBBO paradigms are not tailored to evaluation-limited, expensive settings (Du et al., 2026), often implicitly assuming abundant function evaluations and focusing on *how to search* rather than *where to search*.

In expensive BBO, ineffective exploration can rapidly deplete the evaluation budget, making it crucial to identify and prioritize promising regions (Yang et al., 2017; Pan et al., 2021; Li et al., 2022c). This perspective is consistent

with surrogate-based optimizers, such as Surrogate-Assisted Evolutionary Algorithms (SAEAs), which improve query efficiency by substituting most costly evaluations with surrogate predictions and selecting the next samples via an infill criterion under a tight budget (Li et al., 2022a; Liang et al., 2024). However, such sampling rules are typically heuristic, hand-crafted, and often problem-specific, which limits their adaptability and transferability (Du et al., 2026; Ma et al., 2025d). This motivates combining MetaBBO with surrogate-based optimization, where a meta-policy provides adaptive, data-driven guidance on where to sample in expensive settings. Yet, meta-guided search for surrogate-based optimization remains largely underexplored, primarily due to several fundamental challenges:

1) **Compact and problem-agnostic region abstraction.** A key challenge is to construct a region representation that is compact enough for stable meta-policy learning and problem-agnostic enough for cross-task generalization, while retaining a well-defined ranking of search priority to enable effective guidance.

2) **Effective guidance without excessive restriction.** The challenge is how to guide the underlying optimizer to steer sampling toward selected regions for improved query efficiency, while preserving its intrinsic dynamics and avoiding overly constraining control.

3) **ELA representation for search guidance.** Existing work rarely develops ELA representations tailored to region-level guidance (Mersmann et al., 2011; Malan, 2021; Nunes et al., 2021). In expensive BBO, where landscape information must be inferred from sparse evaluations, the lack of such representations makes it difficult for a meta-policy to reliably guide the search and generalize across tasks.

To enable effective search guidance in MetaBBO, we propose MetaSG-SAEA, a novel MetaBBO framework in which an SAEA serves as the low-level optimizer, while a meta-learned policy guides the SAEA's search for expensive constrained multi-objective optimization problems (ECMOPs). MetaSG-SAEA is designed to address the aforementioned challenges by integrating the following key techniques: (i) We introduce the **Max–Min Constraint-Calibrated Inequality (MM-CCI)**, which maps each solution's constraint evaluations to a compact, problem-agnostic scalar region level in $[0, 1]$, simplifying region partitioning and enabling the meta-policy to efficiently learn search guidance without relying on a complex control space. Additionally, we provide a theoretical analysis and proof of MM-CCI. (ii) To implement region-level guidance, we employ **diffusion-model-based population initialization** to warm-start the SAEA, steering the search toward the target region while preserving its intrinsic evolutionary dynamics without imposing tight constraints. (iii) We replace hand-crafted ELA with an **attention-based representation** that

aggregates objective values and region-level signals from MM-CCI. This design scales across different problem dimensions, population sizes, and numbers of objectives and constraints. In conclusion, our research offers a new perspective in the MetaBBO field, focusing on region-level search guidance, while providing an effective approach for tackling ECMOPs.

## 2. Related Works

### 2.1. Meta-Black-Box-Optimization

MetaBBO views algorithm design as a meta-level learning problem over a distribution of optimization tasks (Zhan et al., 2025; Ma et al., 2025b). Given a set of problem instances sampled from a distribution $\mathcal{P}$, a meta-level policy $\pi_\theta$ is trained to guide a low-level black-box optimizer $\mathcal{O}$ based on its observed optimization states. The objective of MetaBBO is to learn a policy that maximizes the expected performance improvement of the optimizer across tasks (Chen et al., 2024; Guo et al., 2024a; Li et al., 2024). Formally, the meta-objective is defined as

$$J(\theta) = \mathbb{E}_{f\sim\mathcal{P}}[R(\mathcal{O}, \pi_\theta, f)] \approx \frac{1}{N}\sum_{i=1}^{N}\sum_{t=1}^{T}\mathrm{perf}(\mathcal{O}, a_{i,t}, f_i),$$
(1)

where $N$ is the total number of training tasks, and $R(\cdot)$ represents total reward accumulated by following the meta-policy, $a_{i,t} = \pi_\theta(s_{i,t})$ denotes the algorithm design decision produced by the meta-policy from the optimization state $s_{i,t}$ at step $t$, and $s_{i,t}$ is obtained via ELA, i.e., $s_{i,t} = \Lambda_\theta(\mathcal{H}_i^t)$, which maps the available optimization history $\mathcal{H}_i^t$ into a compact state representation (Guo et al., 2025). By maximizing this meta-objective, MetaBBO seeks an optimal meta-policy that improves the performance of the underlying optimizer over a distribution of optimization problems, rather than adapting to a single instance (Liu et al., 2024a).

Existing MetaBBO methods can be broadly categorized by the control space of the meta-policy, i.e., which component of the low-level optimization process is learned and adapted. AS learns to pick a solver from a candidate pool given the current optimization state (Cenikj et al., 2024; Wu et al., 2024b). AC which dynamically adjusts hyperparameters or operators (Xu et al., 2025; Du et al., 2026). SM operates directly in the solution space by generating, perturbing, or refining solutions, thereby partially or even fully replacing the role of the underlying optimizer in proposing new solutions (Yang et al., 2023; Lange et al., 2024). AG explores an open-ended control space to synthesize novel algorithmic workflows (Ma et al., 2024; Liu et al., 2024b). These categories differ in expressiveness and control granularity, reflecting distinct meta-control designs. To clarify our where-to-search positioning, **Appendix E** distinguishes our method from SM.

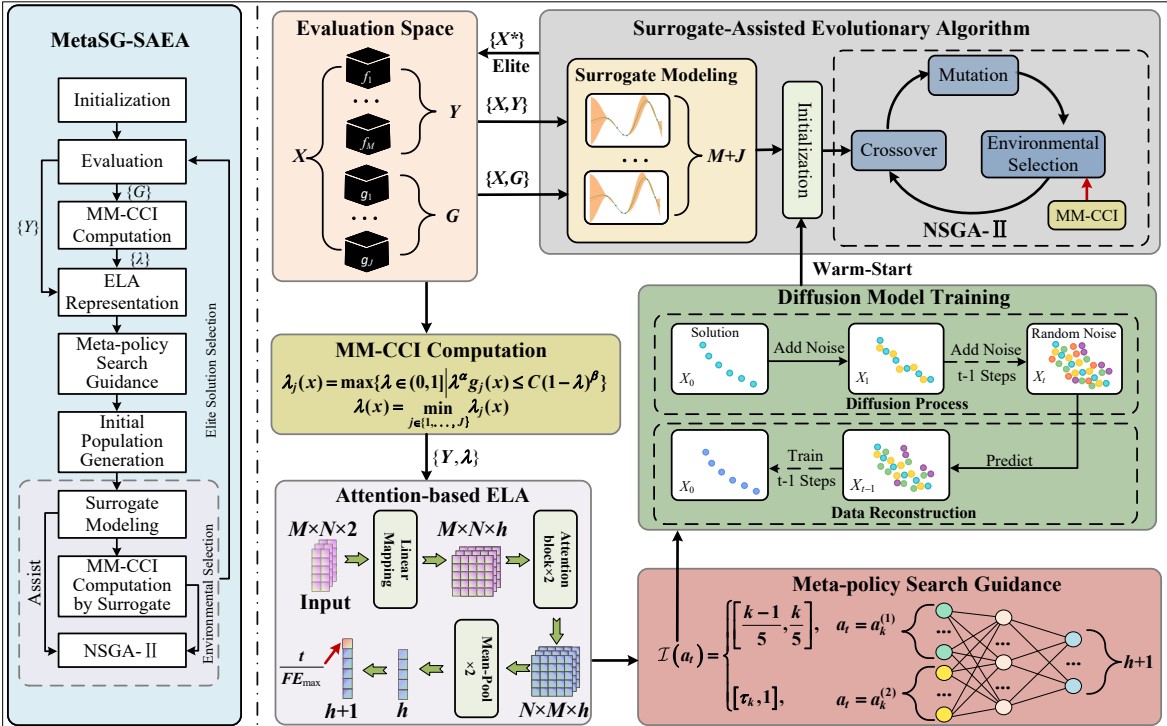

*Figure 2.* MetaSG-SAEA search guidance process. (left) The framework of MetaSG-SAEA. (right) The Meta-policy takes objective values and computed MM-CCI levels as input to determine the search regions. The evaluated solutions within the decision region are passed to the diffusion model for training, which generates the initial population. During the environmental selection phase of the evolutionary algorithm, individuals are selected based on the MM-CCI priority predicted by the surrogate model. This process effectively implements Search Guidance for SAEA.

## 2.2. Surrogate-Assisted Evolutionary Algorithm

SAEAs are widely used for solving ECMOPs. These methods typically replace real-space evaluations with surrogate models in the surrogate space, reducing the number of evaluations required (Liang et al., 2024). Most research in this area can be categorized into two main approaches: one focuses on the design of infill criteria, which guide the search process by selecting the next sample based on the surrogate model's predictions. This includes balancing exploration and exploitation (Singh et al., 2014; Hussein & Deb, 2016; Zhan et al., 2017), as well as integrating multiple infill criteria to improve search efficiency and robustness (Wu et al., 2024a; Zhang et al., 2025). The other approach focuses on the development of stage-wise search strategies, where the optimization process is divided into multiple stages, allowing for a more systematic and adaptive exploration of the search space (Li et al., 2022b; Zhang et al., 2024).

The integration of MetaBBO with SAEA is evolving. For example, DB-SAEA (Du et al., 2026) improves SAEA by adaptively controlling both the infill criterion and the number of evolutionary iterations. DRL-SAEA (Shao et al., 2025) divides the optimization process into multiple stages, with the meta-policy adaptively switching between them. Additionally, DRLOS (Ming et al., 2024) selects operators

using deep reinforcement learning to dynamically adjust the search process, improving flexibility and performance across different tasks. However, as mentioned earlier, the guidance of search regions for expensive optimization problems is a critical aspect, yet it has been inadequately addressed in existing research.

## 3. Methodology

### 3.1. Overview

MetaSG-SAEA follows a bi-level architecture: a meta-policy decides which region to prioritize, while an SAEA performs sample-efficient search under a tight evaluation budget. Formally, we consider an expensive constrained multi-objective optimization problem over a bounded search space $\mathcal{X} \subseteq \mathbb{R}^d$, formulated as:

$$\min_{\mathbf{x} \in \mathcal{X}} \quad \mathbf{F}(\mathbf{x}) = \big(f_1(\mathbf{x}), \dots, f_M(\mathbf{x})\big) \tag{2}$$
$$\text{s.t.} \quad g_j(\mathbf{x}) \leq 0, \quad j = 1, \dots, J.$$

We start from an initial evaluated set generated by Latin Hypercube Sampling (LHS). At each iteration, we apply MM-CCI to map each solution's constraint evaluations to a compact region-level scalar in $[0, 1]$, enabling ordered region partitioning. Based on objective values and the re-

**Algorithm 1** MetaSG-SAEA

---

**Input:** Expensive objectives $\{f_i(\cdot)\}_{i=1}^M$ and constraints $\{g_j(\cdot)\}_{j=1}^J$, Evaluation budget $FE_{\max}$, ELA encoder $\Lambda_\theta$, Diffusion model $\mathcal{M}$, Low-level SAEA optimizer $\mathcal{O}$, Meta-policy $\pi_\theta$.
**Output:** Expensive evaluated solutions.
$X \leftarrow$ Get $N$ initial inputs by LHS;
$Y, G \leftarrow$ Objective and constraint evaluation;
$\lambda \leftarrow$ MM-CCI$(G)$;
Set population $P_0 \leftarrow \{X, Y, G, \lambda\}$;
Set evaluation counter $t \leftarrow N$.
**while** $t < FE_{\max}$ **do**
    Compute state $\boldsymbol{s}_t \leftarrow [\Lambda_\theta(Y, \lambda), t/FE_{\max}]$;
    Obtain region action $a_t \leftarrow \pi_\theta(\boldsymbol{s}_t)$;
    Select evaluated subset $X' \leftarrow Select(a_t, P_t)$;
    $\mathcal{M}' \leftarrow Train(\mathcal{M}, X')$;
    Generate initial population $P_{init} \leftarrow \mathcal{M}'$;
    Elite solution $X^* \leftarrow \mathcal{O}(P_{init})$;
    $Y^*, G^* \leftarrow \{f_i(X^*)\}_{i=1}^M, \{g_j(X^*)\}_{j=1}^J$;
    $\lambda^* \leftarrow$ MM-CCI$(G^*)$;
    $P_t \leftarrow P_t \cup \{X^*, Y^*, G^*, \lambda^*\}$;
    $t \leftarrow t + |X^*|$.
**end while**

---

sulting region signals, we construct an attention-based ELA representation and input the extracted optimization state into the meta-policy to choose a target region. To realize region-level guidance, we train a diffusion model on evaluated solutions filtered by the selected region and sample an informed initial population to warm-start the subsequent SAEA search. During the SAEA's environmental selection, individuals are prioritized according to the surrogate-predicted MM-CCI levels (see Algorithm 1 and Figure 2). This procedure repeats until the evaluation budget is exhausted.

### 3.2. Max–Min Constraint-Calibrated Inequality

In MetaSG-SAEA, we seek a compact scalar signal that summarizes constraint satisfaction and can be used to partition solutions into ordered search regions. We propose MM-CCI, which first assigns a per-constraint calibrated level via **Max-Feasible Constraint-Calibrated Inequality** (Max-CCI) and then aggregates these levels by a worst-constraint rule to obtain an overall region level.

**Max-CCI.** For each constraint $g_j(\mathbf{x}) \leq 0$, we define its calibrated level $\lambda_j(\mathbf{x}) \in (0, 1]$ as the maximum value $\lambda$ that satisfies the **Constraint-Calibrated Inequality** (CCI). Formally:

$$\lambda_j(\mathbf{x}) = \max \left\{ \lambda \in (0, 1] \mid \lambda^\alpha g_j(\mathbf{x}) \leq C(1 - \lambda)^\beta \right\}. \quad (3)$$

where $C > 0$, $\alpha \geq 1$, and $\beta \geq 1$ are hyperparameters. $C$ sets the overall calibration scale. Intuitively, $\alpha$ and $\beta$ adjust

the relative scaling between the violation term on the left-hand side and the allowance term on the right-hand side of the CCI, thereby shaping the mapping from constraint evaluations $g_j(\mathbf{x})$ to the calibrated level $\lambda_j(\mathbf{x})$. Together, these parameters provide flexible control over both the strength and granularity of penalization for infeasible solutions. The proposed Max-CCI formulation enjoys several desirable mathematical properties, as summarized below (with proofs in **Appendix A**):

- **Existence and well-definedness**. For any solution $\mathbf{x}$, there exists at least one $\lambda \in (0, 1]$ that satisfies the CCI. Therefore, the feasible set of $\lambda$ is non-empty, and the calibrated level $\lambda_j(\mathbf{x})$ is well-defined.

- **Monotonicity and boundary consistency**. The Max-CCI level $\lambda_j(\mathbf{x})$ is monotonically non-increasing with respect to the constraint evaluation $g_j(\mathbf{x})$, and is strictly decreasing on the infeasible side ($g_j(\mathbf{x}) > 0$). In particular, when $g_j(\mathbf{x}) \leq 0$, we have $\lambda_j(\mathbf{x}) = 1$, exactly recovering the original feasibility condition. Moreover, larger violations yield smaller calibrated levels, providing a boundary-consistent measure of violation severity.

- **Convexity and diminishing decrease**. On the infeasible side, the Max-CCI level $\lambda_j(\mathbf{x})$ is decreasing and convex with respect to the violation magnitude $g_j(\mathbf{x})$: it drops rapidly for mild violations near the boundary and becomes less steep as violations grow, approaching 0 as $g_j(\mathbf{x}) \to \infty$. This behavior penalizes early infeasibility while preserving resolution among heavily violated solutions, which benefits region-level search guidance in constrained optimization.

In most challenging constrained optimization scenarios, the feasible region typically occupies only a small fraction of the search space, and thus most early evaluations fall into the infeasible region. Due to the diminishing-decrease mapping induced by Max-CCI, infeasible solutions are stratified into pyramid-shaped hierarchical tiers according to their violation severity: severely violated solutions form a wide base, while solution layers become increasingly sparse as they approach the feasibility boundary. This structure enables a smooth transition of the search process from highly infeasible regions to near-feasible ones, allowing the optimizer to incrementally approach feasibility under a strict evaluation budget (see Figure 3 (a)).

**Adaptive hyperparameter estimation**. Considering that CCI involves several hyperparameters, we estimate them in a constraint-wise, data-driven manner using the initial LHS samples. Specifically, for each constraint, we anchor several violation quantiles to pre-defined target MM-CCI levels and solve a lightweight log-linear fitting problem to obtain

---

**Algorithm 2** MM-CCI

---

**Input:** Partition number $K$, solution $\mathbf{x}$ with $\{g_j(\mathbf{x}), \alpha_j, \beta_j, C_j\}_{j=1}^J$
**Output:** MM-CCI level
Set $\Delta \leftarrow 1/K$
**for** $t = 0$ to $K$ **do**
$\quad \lambda_{(t)} \leftarrow 1 - t \cdot \Delta$
$\quad$ **if** $\lambda_{(t)}^{\alpha_j} g_j(\mathbf{x}) \leq C_j (1 - \lambda_{(t)})^{\beta_j}, \forall j \in \{1, \ldots, J\}$ **then**
$\quad\quad$ **Return** $\lambda_{(t)}$
$\quad$ **end if**
**end for**

---

$\{\alpha_j, \beta_j, C_j\}$, which adapts the mapping to heterogeneous constraint scales. Implementation details are deferred to **Appendix B**.

**Worst-constraint aggregation**. For multiple constraints, we adopt a worst-constraint aggregation since feasibility requires satisfying all constraints, making the overall level naturally determined by the bottleneck (least satisfied) one under per-constraint Max-CCI. In contrast, averaging can dilute bottleneck violations and flatten the induced hierarchy, weakening the pyramid-shaped stratification needed for region-level guidance. Specifically, the MM-CCI region level $\lambda(\mathbf{x})$ is defined as:

$$\lambda(\mathbf{x}) = \min_{j \in \{1, \ldots, J\}} \lambda_j(\mathbf{x}). \tag{4}$$

In practice, we compute $\lambda(\mathbf{x})$ on a uniform grid with step size $1/K$ (equivalently, a uniform partition of $(0, 1]$ into $K$ intervals) by scanning candidate $\lambda$ values from 1 down to 0 to approximate the maximal feasible level. This discretized procedure yields a compact and ordered region representation and is summarized in Algorithm 2. Importantly, the resulting discrete levels substantially improve robustness, especially for surrogate-prediction-based environmental selection in SAEA, by stabilizing the ranking across feasibility strata.

Notably, the resulting region levels are problem-agnostic, as they depend only on normalized constraint evaluations through MM-CCI rather than on problem-specific scales, constraint types, or handcrafted thresholds.

### 3.3. Search Guidance for SAEA

**Diffusion-model-based population initialization**. To translate the meta-policy's region decision into solution-level guidance under a tight evaluation budget, we use a denoising diffusion model to generate an informed initial population, serving as a learned prior for the selected region (Ho et al., 2020; Krishnamoorthy et al., 2023; Yao et al., 2026). At each evaluation cycle, we collect the evaluated solutions in the selected region, denoted by $D_r = \{\mathbf{x}_i\}_{i=1}^{N_r}$, and update the diffusion model using $D_r$. The forward process

gradually corrupts a data point $\mathbf{x}_0$ by a Gaussian Markov chain:

$$q(\mathbf{x}_t \mid \mathbf{x}_{t-1}) = \mathcal{N}(\mathbf{x}_t; \sqrt{\alpha_t}\, \mathbf{x}_{t-1}, (1 - \alpha_t)\mathbf{I}), \tag{5}$$

$$q(\mathbf{x}_t \mid \mathbf{x}_0) = \mathcal{N}(\mathbf{x}_t; \sqrt{\bar{\alpha}_t}\, \mathbf{x}_0, (1 - \bar{\alpha}_t)\mathbf{I}), \tag{6}$$

where $\alpha_t = 1 - \beta_t$ and $\bar{\alpha}_t = \prod_{s=1}^t \alpha_s$. Here $t$ indexes the diffusion step, and $\{\beta_t\}$ is a pre-defined noise schedule. We train a noise predictor $\epsilon_\theta(\mathbf{x}_t, t)$ with the standard denoising objective:

$$\mathcal{L}(\theta) = \mathbb{E}_{\mathbf{x}_0 \sim D_r, t, \boldsymbol{\epsilon}} \left[ \left\| \boldsymbol{\epsilon} - \boldsymbol{\epsilon}_\theta(\mathbf{x}_t, t) \right\|_2^2 \right]. \tag{7}$$

Starting from $\mathbf{x}_T \sim \mathcal{N}(\mathbf{0}, \mathbf{I})$, we generate samples by reversing the diffusion process:

$$\mathbf{x}_{t-1} = \frac{1}{\sqrt{\alpha_t}} \left( \mathbf{x}_t - \frac{1 - \alpha_t}{\sqrt{1 - \bar{\alpha}_t}} \boldsymbol{\epsilon}_\theta(\mathbf{x}_t, t) \right) + \sigma_t \mathbf{z}, \tag{8}$$

where $\mathbf{z} \sim \mathcal{N}(\mathbf{0}, \mathbf{I})$ and $\sigma_t^2 = \tilde{\beta}_t = \frac{1 - \bar{\alpha}_{t-1}}{1 - \bar{\alpha}_t} \beta_t$. The resulting samples $\{\mathbf{x}_0\}$ form an informed initial population $P_{\text{init}}$ for warm-starting SAEA in the meta-selected region.

**SAEA of MetaSG-SAEA**. MetaSG-SAEA adopts a standard surrogate-assisted evolutionary optimization loop as the low-level optimizer. We use NSGA-II (Deb et al., 2002) as the evolutionary backbone and Gaussian Process (GP) as the surrogate model. Since region-level guidance is provided by the meta-policy, we use ND-A (Wu et al., 2024a) as a simple infill criterion for elite batch selection.

In each surrogate-based expensive evaluation cycle, we compute the MM-CCI level of each candidate by applying Algorithm 2 to the surrogate-predicted constraint values. For constraint handling, environmental selection follows a level-first rule: candidates with higher MM-CCI levels are prioritized, and NSGA-II's Pareto ranking and diversity preservation are applied within the same level. The uniform-grid discretization in Algorithm 2 further improves the robustness of MM-CCI ranking under surrogate prediction errors. Overall, this design inherits the level-based prioritization induced by MM-CCI while retaining the intrinsic selection and diversity-preservation behavior of NSGA-II.

### 3.4. Model the Evolution Search Procedure as an MDP

To enable dynamic control over the surrogate-assisted multiobjective optimization process, we model the search as a discrete-time, finite-horizon Markov Decision Process (MDP), defined as $\mathcal{M} = (\mathcal{S}, \mathcal{A}, \mathcal{T}, r, \gamma)$, where $\gamma$ denotes discount factor, and $\mathcal{T}$ is state transition.

**State space and attention-based ELA**. $\mathcal{S}$ denotes the state space that reflects optimization status. At each decision step $t$, given the set of evaluated samples $\{X, Y, G\}$, we construct the state vector:

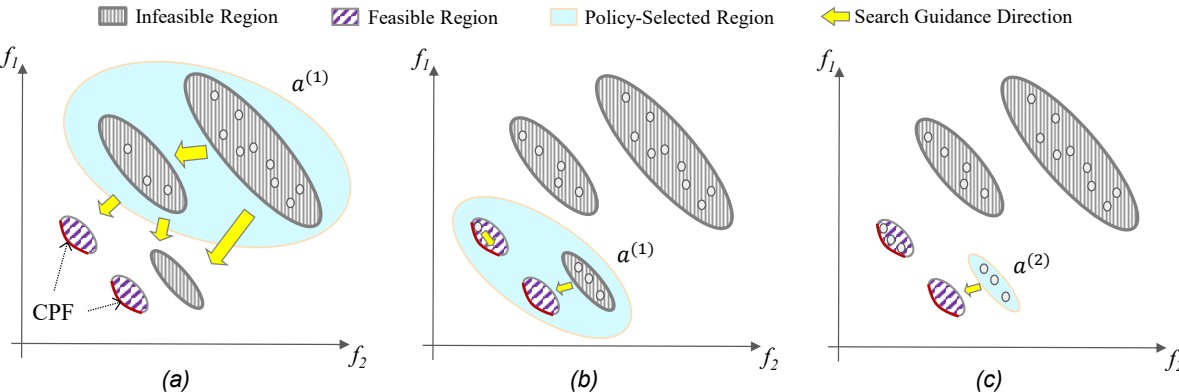

*Figure 3.* Illustration of region-level search guidance. (a) At the early stage, most solutions lie in low-$\lambda$ regions, and the action $a^{(1)}$ provides **broad guidance** by selecting a wide range of levels. (b) As optimization progresses, the guidance shifts toward more diverse near-feasible/feasible solutions. (c) The action $a^{(2)}$ enables more **focused guidance** by restricting selection to a specific MM-CCI level interval. **CPF** denotes the constrained Pareto front.

$$s_t = \left[ \Lambda_\theta(Y, \lambda(X)),\ t/FE_{\max} \right], \quad (9)$$

where $\Lambda_\theta$ is ELA of MetaSG-SAEA, $\lambda(X)$ is the MM-CCI level of $X$, and $t/FE_{max}$ is a normalized scalar indicating the proportion of the evaluation budget that has been consumed up to step $t$. In the ELA module, we first normalize objective values using the observed extrema, and then pair the normalized objectives with the corresponding MM-CCI levels. Specifically, we reorganize the evaluated population into a tensor:

$$E_t = \left\{ \left\{ \left( f'_m(\mathbf{x}_i), \lambda(\mathbf{x}_i) \right) \right\}_{i=1}^{N} \right\}_{m=1}^{M} \in \mathbb{R}^{M \times N \times 2}. \quad (10)$$

We then embed each pair with a linear mapping $W_{emb} \in \mathbb{R}^{2 \times h}$, resulting in an input representation of shape $M \times N \times h$, where $h$ is the hidden dimension. Next, we apply two *Attn* blocks based on the Transformer architecture (Vaswani et al., 2017) with LayerNorm (Ba et al., 2016). The first *Attn* block performs attention across the population to aggregate information among different solutions at aligned dimensions, while the second *Attn* block performs attention within each solution to further capture interactions across different dimensions. Finally, mean pooling is applied over the first two dimensions, yielding an $h$-dimensional representation of the optimization state. This design yields scalable policy inputs across tasks with varying problem dimensions, population sizes, and numbers of objectives and constraints. Additional technical details can be found in **Appendix C**.

Different from prior attention-based ELA methods that embed decision variables (Ma et al., 2025a; Du et al., 2026), we do not include $X$ in the ELA input. Our search guidance is formulated at the objective and feasibility levels rather than in the decision space: the meta-policy only needs signals that describe performance trade-offs and feasibility progres-

sion (see Figure 3). Accordingly, the objective values $Y$ together with the MM-CCI levels provide concise yet sufficient information for effective guidance, while avoiding the problem-dependent dimensionality and scaling issues of $X$, which can hinder transfer across heterogeneous tasks.

**Action space.** MetaSG-SAEA considers two categories of discrete actions, forming a discrete action space with 10 actions, $\mathcal{A} = \{a_k^{(1)}, a_k^{(2)} \mid k = 1, \cdots, 5\}$, to control training-set selection for the diffusion model. Specifically, each action $a_t \in \mathcal{A}$ corresponds to a predefined MM-CCI level interval, denoted by $\mathcal{I}(a_t)$, and the evaluated samples whose MM-CCI levels fall within this interval are selected to construct the diffusion model's training set. The region induced by $a_t$ is defined as:

$$\mathcal{I}(a_t) = \begin{cases} [\tau_k, 1], & a_t = a_k^{(1)}, \\[2mm] \left[ \dfrac{k-1}{5}, \dfrac{k}{5} \right], & a_t = a_k^{(2)}, \end{cases} \quad (11)$$

where $\tau_k \in \{0,\ 0.45,\ 0.70,\ 0.90,\ 1\}$. Selecting different MM-CCI level regions determines the direction of search guidance. This design therefore provides diverse guidance directions, as illustrated in Figure 3. If the selected region does not contain a sufficient number of samples (set to 20 in our experiments), we expand the range by including samples from nearby levels closest to the target interval.

**Reward function.** We define the reward to reflect the immediate progress after executing an action and evaluating the selected solutions. Specifically, it combines the increment of the population's maximum MM-CCI level and the relative Inverted Generational Distance (IGD) improvement:

$$r_t = \left( \max_{\mathbf{x} \in P_t} \lambda(\mathbf{x}) - \max_{\mathbf{x} \in P_{t-1}} \lambda(\mathbf{x}) \right) + \frac{\text{IGD}_{t-1} - \text{IGD}_t}{\text{IGD}_{t-1}}. \quad (12)$$

### 3.5. Meta-Policy Training

We adopt a parallel sampling and centralized training paradigm (Du et al., 2026) based on an off-policy Double DQN framework (Van Hasselt et al., 2016), where the target network is updated via soft updates (Haarnoja et al., 2018). Multiple ECMOP environments interact in parallel with the current meta-policy to generate heterogeneous state–action–reward transitions. These experiences are aggregated into a centralized replay buffer and used to jointly train a shared-parameter meta-policy. This sampling–training loop is repeated until performance stabilizes. Notably, the ELA module is co-trained end-to-end with the meta-policy during reinforcement learning. To enable parallel sampling, we employ **Ray** (Moritz et al., 2018), an open-source framework for parallel processing in machine learning applications. With Ray, the sampling tasks can be distributed across multiple CPUs and GPUs, allowing simultaneous interaction with multiple environments.

## 4. Experimental Studies

In this section, we aim to address the following research questions: **RQ1**: How stable and convergent is meta-policy training across multiple environments? **RQ2**: How well does the proposed approach generalize to unseen tasks? **RQ3**: Do different meta-policies learn consistent search guidance for problem distributions? **RQ4**: Is MM-CCI necessary for guiding the search process? Additional experimental results are provided in **Appendix D**.

### 4.1. Experimental Setup

**Benchmark problems and baselines**. We conduct experiments on the MW (Ma & Wang, 2019) and DAS-CMOP (Fan et al., 2020) benchmark suites, which provide diverse constrained multi-objective landscapes with varying feasible-region complexity and difficulty levels. Moreover, we compare MetaSG-SAEA with state-of-the-art SAEA-based methods (KTS (Song et al., 2021), EIC-MSSAEA (Wu et al., 2024a)) and MetaBBO-based approaches (DR-LOS (Ming et al., 2024), DRL-SAEA (Shao et al., 2025)) to evaluate the effectiveness of the proposed search guidance mechanism.

**Experimental settings**. The update rate for the soft target-network mechanism is set to $\tau = 1e^{-4}$. The training batch size is set to 64. We use the Adam optimizer, with an initial learning rate of $1e^{-5}$ for the ELA module and $1e^{-4}$ for the backbone network. The attention blocks use a single-head configuration with a hidden dimension of $h = 16$. For fair comparison, all methods start with 100 LHS samples and share the same evaluation budget, $FE_{\max} = 300$. The evolutionary search runs for 15 generations, with all SAEAs executed for 40 batches, each containing 5 elite solutions

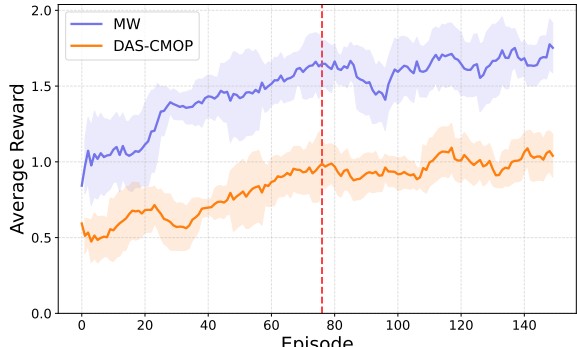

*Figure 4.* Average reward during meta-policy training.

selected for expensive evaluation, across all algorithms. Additionally, the value of $K$ for MM-CCI is set to 40. Further training and testing details are provided in the following experimental sections. All experiments were conducted on a compute cluster equipped with 14 × 32GB VGPUs and 14 × 25-core Intel Xeon Platinum 8481C processors.

**Evaluation metric**. We adopt IGD as the primary metric, computed on the feasible Pareto front to evaluate the convergence and diversity of feasible solutions.

### 4.2. Training Stability and Convergence (RQ1)

To examine the training stability of the proposed meta-policy, we separately train two models on the MW and DAS-CMOP benchmark suites, respectively, as illustrated in Figure 4. In each setting, the meta-policy is trained by interacting in parallel with all problems in the corresponding suite at dimension $d = 8$, including 14 MW problems and 9 DAS-CMOP problems. To reduce stochastic fluctuations, we report the evolution of the average reward computed across environments using a sliding-window average with a window size of 10. As shown in Figure 4, both training processes exhibit a clear convergence trend, with the average reward stabilizing after the red dashed vertical line. The shaded regions indicate ±1 standard deviation computed from 10 independent runs of the smoothed reward.

### 4.3. Zero-shot Performance (RQ2)

To evaluate zero-shot generalization, we perform cross-suite and cross-dimensional transfer. Two meta-policies are trained at $d = 8$, one on MW and the other on DAS-CMOP. Without fine-tuning, the MW-trained policy is tested on DAS-CMOP at $d = 10$, and the DAS-CMOP-trained policy is tested on MW at $d = 10$. Hence, each policy is evaluated on unseen problem distributions with a higher decision-space dimension. We compare each baseline with MetaSG-SAEA using the Wilcoxon rank sum test (Derrac et al., 2011) at the 0.05 significance level over 15 independent runs. Table 1 shows that MetaSG-SAEA generally achieves better performance under 300 evaluations. "NaN"

*Table 1.* Cross-suite and cross-dimensional zero-shot transfer performance (mean (std)).

| PROBLEM | KTS | EIC-MSSAEA | DRLOS | DRL-SAEA | MetaSG-SAEA |
|---------|-----|------------|-------|----------|-------------|
| MW1 | 4.519e-1 (2.12e-1)- | 6.756e-1 (8.70e-2)- | NaN (NaN) | NaN (NaN) | **3.892e-1 (9.95e-2)** |
| MW2 | 3.801e-1 (2.30e-1)- | 3.005e-1 (2.07e-1)- | NaN (NaN) | 2.977e-1 (1.51e-1)- | **1.415e-1 (1.05e-1)** |
| MW3 | 1.011e-1 (1.49e-2)- | 9.658e-2 (1.04e-2)- | 2.33e-1 (1.33e-1)- | 8.678e-2 (9.38e-3)= | **8.320e-2 (1.24e-2)** |
| MW4 | NaN (NaN) | 3.831e-1 (1.04e-1)= | NaN (NaN) | NaN (NaN) | **3.769e-1 (7.09e-2)** |
| MW5 | NaN (NaN) | NaN (NaN) | NaN (NaN) | NaN (NaN) | **5.783e-1 (2.27e-1)** |
| MW6 | 7.390e-1 (3.81e-1)- | 7.417e-1 (2.89e-1)- | NaN (NaN) | 6.233e-1 (3.50e-1)- | **5.880e-1 (2.84e-1)** |
| MW7 | 8.821e-2 (2.90e-2)- | 7.866e-2 (2.05e-2)- | 2.458e-1 (2.10e-1)- | 7.751e-2 (7.44e-3)- | **6.058e-2 (8.38e-3)** |
| MW8 | 8.724e-1 (1.69e-1)- | 5.553e-1 (2.23e-1)- | 7.532e-1 (2.10e-1)- | 3.886e-1 (9.04e-2)- | **3.344e-1 (7.05e-2)** |
| MW9 | 5.893e-1 (2.27e-1)- | 5.039e-1 (2.44e-1)- | NaN (NaN) | NaN (NaN) | **4.906e-1 (4.12e-2)** |
| MW10 | NaN (NaN) | NaN (NaN) | NaN (NaN) | NaN (NaN) | **4.276e-1 (1.65e-1)** |
| MW11 | 1.440e-1 (6.57e-2)+ | 1.230e-1 (3.49e-2)+ | 9.540e-1 (5.25e-2)- | **1.032e-1 (2.02e-2)+** | 8.070e-1 (1.58e-1) |
| MW12 | 8.961e-1 (3.00e-1)- | 7.961e-1 (2.03e-1)- | NaN (NaN) | 7.442e-1 (1.55e-1)- | **6.035e-1 (2.58e-1)** |
| MW13 | 1.562e+0 (8.07e-1)- | 1.184e+0 (4.43e-1)- | 2.104e+0 (1.61e+0)- | 1.212e+0 (5.35e-1)- | **1.003e+0 (4.03e-1)** |
| MW14 | 1.425e+0 (6.71e-1)- | 1.549e+0 (4.23e-1)- | 3.345e+0 (5.88e-1)- | 1.616e+0 (3.01e-1)- | **1.264e+0 (4.07e-1)** |
| DAS-CMOP1 | 6.359e-1 (2.53e-1)- | 2.992e-1 (6.23e-2)- | 6.312e-1 (1.76e-1)- | 3.847e-1 (1.05e-1)- | **1.893e-1 (3.31e-2)** |
| DAS-CMOP2 | 2.454e-1 (1.82e-2)- | 1.962e-1 (4.40e-2)- | 4.542e-1 (1.11e-1)- | 1.588e-1 (1.33e-2)- | **1.108e-1 (2.58e-2)** |
| DAS-CMOP3 | 5.195e-1 (1.00e-2)- | 4.794e-1 (2.50e-2)- | 5.772e-1 (1.32e-1)- | 3.253e-1 (2.35e-2)- | **2.403e-1 (5.10e-2)** |
| DAS-CMOP4 | NaN (NaN) | 4.596e-1 (2.92e-1)+ | NaN (NaN) | **4.190e-1 (1.20e-1)+** | NaN (NaN) |
| DAS-CMOP5 | NaN (NaN) | **4.091e-1 (5.01e-2)+** | NaN (NaN) | 5.201e-1 (4.23e-2)+ | NaN (NaN) |
| DAS-CMOP6 | 5.133e-1 (8.17e-2)+ | **4.921e-1 (1.02e-1)+** | NaN (NaN) | 5.339e-1 (5.19e-2)+ | NaN (NaN) |
| DAS-CMOP7 | 7.973e-1 (2.52e-1)- | 5.607e-1 (1.47e-1)= | 1.824e+0 (5.05e-1)- | 6.329e-1 (1.99e-1)- | **5.479e-1 (2.61e-1)** |
| DAS-CMOP8 | 1.077e+0 (5.19e-1)- | 5.520e-1 (1.54e-1)- | 3.394e+0 (1.20e+0)- | 6.450e-1 (1.23e-1)- | **4.486e-1 (2.09e-1)** |
| DAS-CMOP9 | 4.299e-1 (7.23e-2)- | 3.424e-1 (1.22e-1)- | 7.232e-1 (5.52e-2)- | 4.428e-1 (9.82e-2)- | **2.682e-1 (7.02e-2)** |
| + / - / = | 2 / 21 / 0 | 4 / 17 / 2 | 0 / 23 / 0 | 4 / 18 / 1 | |

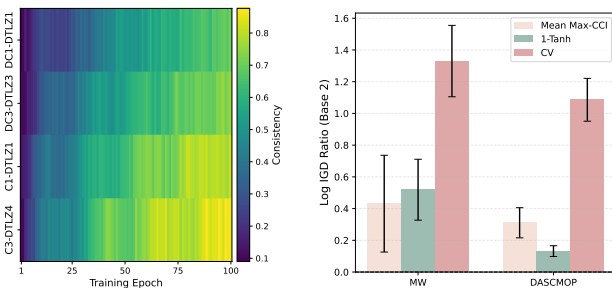

*Figure 5.* (left) Decision consistency heatmap across meta-policies over training. (right) MM-CCI comparison with other methods.

denotes that no feasible solution is found within the budget. The failure cases on DAS-CMOP4–6 are discussed in **Appendix D.4**.

### 4.4. Consistency and MM-CCI Comparison (RQ3, RQ4)

We evaluate whether MetaSG-SAEA learns search guidance tailored to problem distributions by measuring the decision consistency between two meta-policies trained on the MW and DAS-CMOP suites. At each training checkpoint, both policies are deployed on the same external test problems under identical optimization states. The test problems include DC-DTLZ and C-DTLZ variants (Deb & Jain, 2013). After identical initial sampling, the remaining 200 evaluations are used for guidance, resulting in 40 decisions (5 samples per cycle). Consistency is defined as the fraction of steps where both policies select the same action. As shown in Figure 5 (left), this consistency increases over training, indi-

cating that the two meta-policies converge to more aligned guidance behaviors under the problem distribution.

Figure 5 (right) shows a comparison of MM-CCI with the mean-aggregated Max-CCI, as well as the performance of the constraint using the *1-Tanh mapping* and the method without the search guidance Constraint Violation (CV) criterion (Qiao et al., 2024). The *1-Tanh mapping* is used to transform the constraint violation to the range $[0, 1]$. This method shares some properties with Max-CCI, but it lacks the flexibility to adjust the mapping shape according to the specific problem. If a feasible solution is not found within 300 evaluations, the search continues until one is found, and IGD is then computed. The results highlight the necessity of the MM-CCI design and the need for search guidance.

## 5. Conclusion

In this paper, we introduce MetaSG-SAEA, which differs from most MetaBBO methods by not affecting the optimizer internals, but instead learning an external search guidance policy. Experimentally, this guidance is particularly effective for ECMOPs, significantly improving optimization performance under tight evaluation budgets. Future work includes: (i) integrating internal control of the optimizer, such as operator selection and infill-criterion selection, with external guidance, potentially via hierarchical reinforcement learning or multi-agent formulations; (ii) extending the framework beyond multi-objective settings to broader black-box optimization scenarios to enhance generality.

## Acknowledgements

This work was supported in part by the National Natural Science Foundation of China under Grants 72301286, 72431011 and 62303474.

## Impact Statement

This research advances the field of evolutionary computation and Meta-Black-Box Optimization by offering a new perspective focused on the selection of search regions, rather than the traditional focus on search strategies. By introducing a region-level search guidance strategy, this work provides an innovative approach to tackling expensive constrained multi-objective optimization problems. Unlike existing methods that primarily optimize the search process, our contribution lies in optimizing the search space itself, enhancing efficiency, reducing resource consumption, and advancing applications in industrial design, resource allocation, and scientific discovery. This new perspective is expected to accelerate the development of automated algorithm design and promote the broader application of optimization technologies.

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

# A. Theoretical Analysis of Max-CCI

In this appendix, we present the theoretical analysis of the proposed Max-CCI. We establish the fundamental mathematical properties of the per-constraint calibrated level, including existence and well-definedness, monotonicity and boundary consistency with respect to constraint violation, as well as the diminishing-contraction behavior induced by calibrated mapping. These results provide a rigorous foundation for the interval stratification induced by Max-CCI and support its use as a compact, ordered representation of constraint satisfaction.

## A.1. Existence and well-definedness

We first state a basic property of the proposed Max-CCI level, which ensures that the definition is meaningful for any evaluated solution. For any evaluated solution $\mathbf{x}$ and any constraint $g_j(\mathbf{x})$, there exists at least one $\lambda \in (0, 1]$ satisfying the CCI. As a result, the per-constraint Max-CCI level $\lambda_j(\mathbf{x})$, defined as the maximum feasible $\lambda$, is well-defined. This property follows directly from the structure of the CCI and can be seen by considering two simple cases:

If $g_j(\mathbf{x}) \leq 0$, then $\lambda^\alpha g_j(\mathbf{x}) \leq 0$ holds for all $\lambda \in (0, 1]$, while the right-hand side $C(1 - \lambda)^\beta$ is non-negative. Therefore, the CCI is satisfied for the entire interval $(0, 1]$, and the maximal feasible level is $\lambda_j(\mathbf{x}) = 1$. If $g_j(\mathbf{x}) > 0$, consider the continuous function on $\lambda \in [0, 1]$:

$$\phi(\lambda) = C(1 - \lambda)^\beta - \lambda^\alpha g_j(x),$$

At $\lambda = 0$, we have $\phi(0) = C > 0$, while at $\lambda = 1$, $\phi(1) = -g_j(\mathbf{x}) < 0$. By continuity, there exists at least one $\lambda^\star \in (0, 1)$ for which the CCI holds. Hence, the feasible set of $\lambda$ is non-empty and bounded above by 1, which guarantees the existence of a well-defined maximal value $\lambda_j(\mathbf{x}) \in (0, 1]$.

## A.2. Monotonicity and boundary consistency

We next analyze the monotonicity of the Max-CCI with respect to constraint violation, as well as its consistency with the original constraint boundary.

**Monotonicity**. Recall that $\lambda_j(\mathbf{x})$ is defined as

$$\lambda_j(\mathbf{x}) = \max\left\{\lambda \in (0, 1] : \ \lambda^\alpha g_j(\mathbf{x}) \leq C(1 - \lambda)^\beta\right\},$$

where $C > 0$, $\alpha \geq 1$, and $\beta \geq 1$. If $g_j(\mathbf{x}) \leq 0$, then for any $\lambda \in (0, 1]$, we have $\lambda^\alpha g_j(\mathbf{x}) \leq 0$ while $C(1 - \lambda)^\beta \geq 0$. Hence the inequality is satisfied for all $\lambda \in (0, 1]$, and thus $\lambda_j(\mathbf{x}) \equiv 1$ on the feasible side (monotone invariant). If $g_j(\mathbf{x}) > 0$, let $v := g_j(\mathbf{x})$ and define

$$\mathcal{S}(v) := \left\{\lambda \in (0, 1] : \ v\lambda^\alpha \leq C(1 - \lambda)^\beta\right\}.$$

For $0 < v_1 < v_2$, we have $\mathcal{S}(v_2) \subseteq \mathcal{S}(v_1)$, since any $\lambda$ feasible under $v_2$ is also feasible under the smaller coefficient $v_1$. Therefore,

$$\lambda^\star(v_2) := \max \mathcal{S}(v_2) \ \leq \ \max \mathcal{S}(v_1) =: \lambda^\star(v_1),$$

implying $\lambda_j(\mathbf{x}) = \lambda^\star(g_j(\mathbf{x}))$ is non-increasing in $g_j(\mathbf{x})$ for $g_j(\mathbf{x}) > 0$. To obtain strict decrease, note that for $v > 0$ the maximizer cannot have slack; hence $\lambda^\star(v) \in (0, 1)$ satisfies

$$v(\lambda^\star)^\alpha = C(1 - \lambda^\star)^\beta.$$

Define $h(\lambda) := \dfrac{C(1 - \lambda)^\beta}{\lambda^\alpha}$ on $(0, 1)$. Then $v = h(\lambda^\star(v))$ and $h'(\lambda) = -C(1-\lambda)^{\beta-1}\lambda^{-\alpha-1}\big(\beta\lambda + \alpha(1 - \lambda)\big) < 0$, so $h$ is strictly decreasing and invertible. Thus $\lambda^\star(v) = h^{-1}(v)$ is strictly decreasing in $v$, i.e., for $0 < v_1 < v_2$, $\lambda^\star(v_1) > \lambda^\star(v_2)$. This proves $\lambda_j(\mathbf{x})$ strictly decreases with violation magnitude on the infeasible side.

**Boundary consistency**. From the feasible-side result above, whenever $g_j(\mathbf{x}) \leq 0$, we have $\lambda_j(\mathbf{x}) = 1$. Hence Max-CCI exactly recovers the original feasibility condition at and inside the constraint boundary.

## A.3. Convexity and diminishing decrease

We analyze the shape of the per-constraint Max-CCI level on the infeasible side. Let $v := g_j(\mathbf{x}) > 0$ denote the violation magnitude. Recall that the corresponding Max-CCI level $\lambda_j(v) \in (0, 1)$ is determined by the boundary condition between feasibility and infeasibility.

Define the contraction amount:
$$c(v) := 1 - \lambda_j(v) \in (0, 1).$$

For $v > 0$, $\lambda_j(v)$ (equivalently $c(v)$) is uniquely determined and satisfies the implicit relation:

$$v = C\, c(v)^\beta \big(1 - c(v)\big)^{-\alpha},$$

where $C > 0$, $\alpha \geq 1$, and $\beta \geq 1$. Define the mapping:

$$q(c) := C\, c^\beta (1 - c)^{-\alpha}, \qquad c \in (0, 1),$$

so that $v = q(c(v))$, i.e., $c(v) = q^{-1}(v)$.

**Step 1 (Monotonicity of $q$).**   We show that $q$ is strictly increasing on $(0, 1)$. Since $q(c) > 0$ for all $c \in (0, 1)$, we compute the log-derivative:
$$\frac{q'(c)}{q(c)} = \frac{\beta}{c} + \frac{\alpha}{1 - c} > 0.$$

Hence $q'(c) > 0$ on $(0, 1)$ and $q$ is strictly increasing. Therefore, for every $v > 0$, the inverse function $c(v) = q^{-1}(v)$ is well-defined and unique.

**Step 2 (Convexity of $q$).**   Differentiating again yields:
$$\frac{q''(c)}{q(c)} = \frac{\beta(\beta - 1)}{c^2} + \frac{2\alpha\beta}{c(1 - c)} + \frac{\alpha(\alpha + 1)}{(1 - c)^2}.$$

Under $\alpha \geq 1$ and $\beta \geq 1$, the right-hand side is strictly positive for all $c \in (0, 1)$. Since $q(c) > 0$ on $(0, 1)$, it follows that $q''(c) > 0$ on $(0, 1)$, and thus $q$ is strictly convex on $(0, 1)$.

**Step 3 (Concavity of the inverse and diminishing contraction).**   A standard property of inverse functions states that the inverse of a strictly increasing convex function is concave. Since $q$ is strictly increasing and strictly convex, its inverse $c(v) = q^{-1}(v)$ is strictly increasing and concave. Consequently, the contraction amount exhibits a diminishing effect: the marginal increase $c'(v)$ decreases as $v$ grows.

**Step 4 (Convexity and diminishing decrease of $\lambda_j$).**   Because $\lambda_j(v) = 1 - c(v)$ and $c(v)$ is concave, $\lambda_j(v)$ is strictly decreasing and convex in $v$. Equivalently, $\lambda_j(v)$ drops rapidly for mild violations near the feasibility boundary and becomes less steep as violations grow (diminishing decrease).

Finally, note that $q(c) \to \infty$ as $c \to 1^-$ (since $(1 - c)^{-\alpha} \to \infty$). Hence $v \to \infty$ implies $c(v) \to 1$, and therefore:

$$\lambda_j(v) = 1 - c(v) \to 0.$$

## B. Adaptive Hyperparameter Estimation for CCI

Considering that CCI involves several hyperparameters, we adopt a constraint-wise, data-driven estimation strategy that relies only on the initial LHS design. For each constraint $g_j(\mathbf{x}) \leq 0$, we compute violation magnitudes on the initial samples:
$$v_j(\mathbf{x}_i) = \max\{0,\, g_j(\mathbf{x}_i)\}.$$

We then keep only samples with $v_j(\mathbf{x}_i) > 0$ when estimating the hyperparameters. We extract a few empirical quantiles $\{\hat{v}_{j,r}\}$ from these positive violations to characterize the typical violation scale of constraint $j$, where $r \in (0, 1)$ denotes the quantile level computed over $\{v_j(\mathbf{x}_i) : v_j(\mathbf{x}_i) > 0\}$.

Next, we pre-specify a set of per-constraint Max-CCI target levels $\{\lambda_r\} \subset (0, 1)$ and anchor each quantile $\hat{v}_{j,r}$ to $\lambda_r$ through the tight form of the Max-CCI mapping on the infeasible side:

$$\hat{v}_{j,r} = C_j\, \frac{(1 - \lambda_r)^{\beta_j}}{(\lambda_r)^{\alpha_j}}.$$

Taking logarithms yields an approximately linear model in $(\log C_j, \beta_j, \alpha_j)$:

$$\log \hat{v}_{j,r} = \log C_j + \beta_j \log(1 - \lambda_r) - \alpha_j \log(\lambda_r),$$

which we solve via a lightweight least-squares fit to obtain $(\alpha_j, \beta_j, C_j)$ for each constraint. This procedure adapts the Max-CCI mapping to heterogeneous constraint scales observed in the initial samples with negligible overhead. When a constraint provides too few positive violations in the initial population, we apply a simple fallback for stability by fixing $\alpha_j = 1$, and $\beta_j = 2$ to default values and estimating only $C_j$ from a single anchor.

**Note.** In this work, we use five quantile anchors $r \in \{0.95, 0.75, 0.50, 0.25, 0.05\}$, paired with per-constraint Max-CCI target levels $\lambda_r \in \{0.01, 0.125, 0.25, 0.375, 0.50\}$.

## C. Technical Details of Attention-based ELA

Within this paper, landscape analysis aims to profile the dynamic optimization status of an ongoing search process, serving as the state feature for meta-level decision making. A neural ELA module should satisfy two requirements: (i) *generalizability* across different search ranges and objective/constraint scales, and (ii) *scalability* to varying population sizes and numbers of objectives/constraints. To this end, we adopt an attention-based ELA that operates on objective values paired with MM-CCI levels and aggregates information via two attention blocks.

**ELA Input Construction**. We normalize objective values using the observed extrema and pair each normalized objective with its MM-CCI level. The evaluated population is organized as $E_t = \{\{(f'_m(\mathbf{x}_i), \lambda(\mathbf{x}_i))\}_{i=1}^N\}_{m=1}^M$, where $\lambda(\mathbf{x}_i)$ denotes the MM-CCI level. Each pair is then projected by a linear embedding $W_{\text{emb}} \in \mathbb{R}^{2 \times h}$ to obtain an input tensor $E'_t \in \mathbb{R}^{M \times N \times h}$.

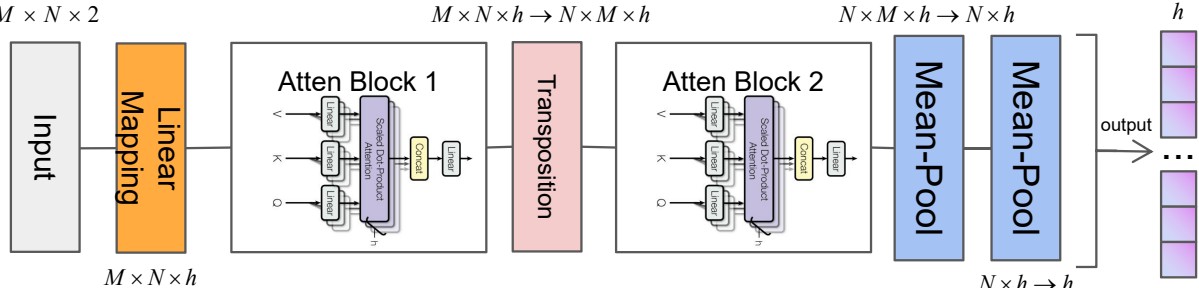

*Figure 6.* The computation graph of attention-based ELA.

**Attention-Based Aggregation**. As illustrated in Figure 6, we employ two attention blocks to aggregate information. Each block follows the standard Transformer design with residual connections and feed-forward layers, but replaces batch normalization with layer normalization to better handle variable population sizes and stabilize training. The first block, denoted as $Attn_1$, performs the following computation:

$$g = \text{LN}\big(E'_t + \text{MHSA}(E'_t)\big),$$
$$v = \text{FF}^{(2)}\Big(\text{ReLU}\big(\text{FF}^{(1)}(g)\big)\Big),$$
$$E''_t = \text{LN}(g + v),$$

where LN denotes layer normalization, MHSA is multi-head self-attention, and FF is a two-layer feed-forward network. The resulting features are then transposed back to $\mathbb{R}^{M \times N \times h}$ for the subsequent aggregation stage. To encode the ordering across objectives/dimensions, we add sinusoidal positional encoding (cos/sin PE) to the transposed tensor $E''_t$. The enhanced representation is fed into the second attention block, $Attn_2$, which promotes information exchange across different objectives/dimensions. Finally, we apply mean pooling over the first two axes to obtain a fixed-length ELA embedding. The

computation is summarized as follows:

$$E_t'' = Attn_1(E_t') \in \mathbb{R}^{M \times N \times h},$$

$$E_t''' = Attn_2\Big(\text{PE}\big(\text{Transpose}(E_t'')\big)\Big) \in \mathbb{R}^{N \times M \times h},$$

$$\boldsymbol{s}_t = \text{MeanPool}_M\Big(\text{MeanPool}_N\big(E_t'''\big)\Big) \in \mathbb{R}^h.$$

# D. Additional Discussion

### D.1. Sensitivity Analysis

In this section, we investigate the sensitivity of MetaSG-SAEA to two key parameters: the elite solution batch size $n_{bs}$ and the number of segments $K$ in MM-CCI. The $n_{bs}$ refers to the number of solutions selected for evaluation in each cycle of the evolutionary process. This parameter controls the granularity of the search and the computational cost. The second parameter, $K$, determines the number of segments in MM-CCI, which plays a critical role in balancing the trade-off between the precision and robustness in the region-level ranking, where larger intervals offer more robustness, and smaller intervals provide finer granularity. We conduct experiments to assess the impact of these parameters on the performance and efficiency of MetaSG-SAEA. Notably, we do not retrain the meta-policy for these parameters, but directly transfer the learned policy to new parameter settings, while also verifying the transferability of MetaSG-SAEA under different parameter configurations.

As shown in Figure 7, small changes in the elite solution batch size $n_{bs}$ and the number of segments $K$ in MM-CCI have little impact on the optimization performance. This suggests that the method is relatively stable with respect to these parameters, and small adjustments do not significantly affect the overall results.

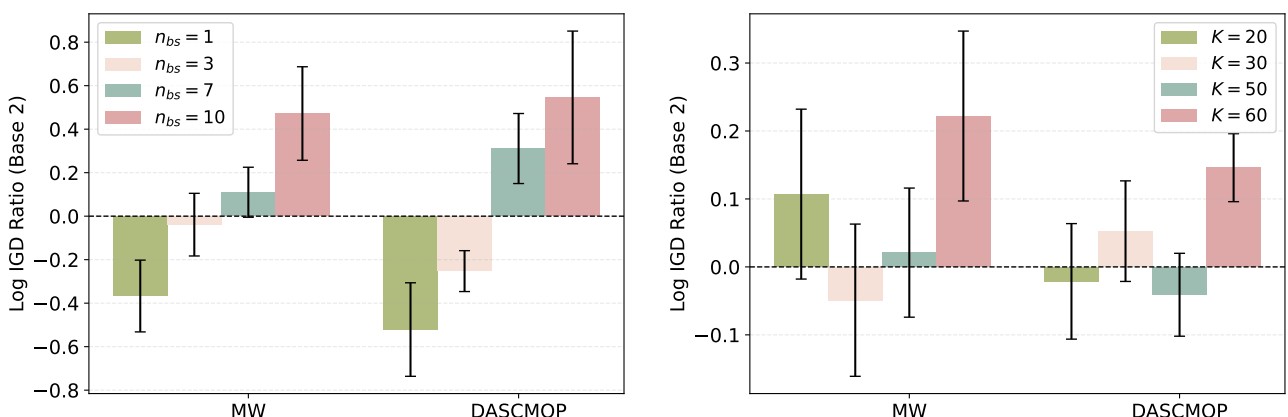

*Figure 7.* Impact of elite solution batch size and MM-CCI segment count on optimization performance.

### D.2. Ablation Study

In this section, we present an ablation study to evaluate the impact of key components in MetaSG-SAEA, including the two types of actions in the action space, the adaptive hyperparameter estimation for Max-CCI, and the diffusion-model-based warm-start for population initialization. For the ablation of the action space, we remove one category of actions and select the action with the highest softmax probability from the remaining ones. For the ablation of the adaptive hyperparameter estimation for Max-CCI, we fix the parameters to $\alpha = 1, \beta = 2$ and $\alpha = 2, \beta = 3$. For the ablation of diffusion warm-start, we disable the diffusion model and instead directly construct the initial population using the evaluated samples that fall within the meta-policy selected MM-CCI interval. By removing or altering these components, we aim to understand their contributions to the overall effectiveness and generalization ability of the method.

As shown in Figure 8, all ablation variants lead to an increase in IGD, resulting in degraded optimization performance. Notably, the impact of ablating $a^{(1)}$ (denoted as '$-a^{(1)}$' in the figure) is more significant than that of $a^{(2)}$ (denoted as '$-a^{(2)}$' in the figure), highlighting the importance of guiding the search direction over a broad range. Moreover, the variant without the diffusion model (denoted as '$-\mathcal{M}$') also exhibits a clear performance drop, demonstrating that diffusion-based

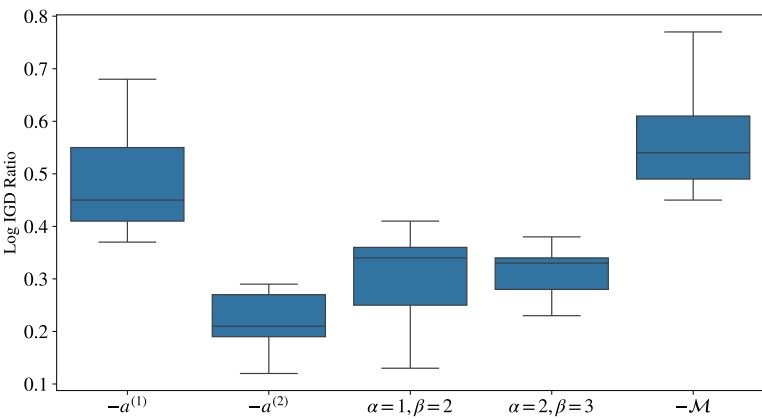

*Figure 8.* Ablation Study of MetaSG-SAEA on MW Problems.

warm-start is necessary for effectively translating the meta-policy's region-level decision into informative solution-level priors and thus improving the overall optimization performance.

### D.3. Model Complexity Comparison

To further analyze the impact of model complexity in MetaSG-SAEA's attention architecture, we compare six configurations with different numbers of attention heads $L \in \{1, 3, 5\}$ and hidden dimensions $h \in \{16, 64\}$. We evaluate the models on two indicators: (i) zero-shot performance on MW test tasks, and (ii) the final average reward achieved during training on DAS-CMOP problems. Results are summarized in a 2D scatter plot, where the y-axis reports the base-2 logarithmic IGD ratio averaged over test problems, and the x-axis shows the base-2 logarithmic ratio of the final training reward relative to the base design ($h = 16, L = 1$). To reduce computational cost, we primarily train the model using a large offline dataset collected in the RQ1 experiments, and only perform a limited number of online interactions after convergence for final refinement.

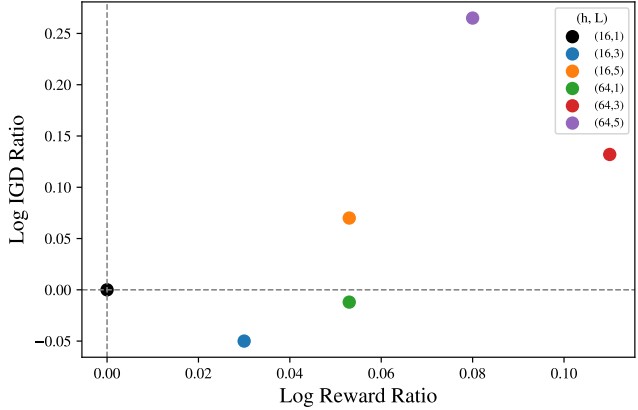

*Figure 9.* The impact of model complexity on zero-shot performance and training reward.

As shown in Figure 9, the experimental results reveal a general trend: as model complexity increases, the training reward tends to improve. However, while a slight increase in complexity can marginally enhance optimization performance, excessive complexity leads to degraded zero-shot performance on unseen test tasks. This can be attributed to two factors: (1) the current model complexity is already sufficient to learn the search guidance for problem distributions, and further complexity offers diminishing returns; (2) the limited number of training environments leads to overfitting in more complex models, reducing their ability to generalize to unseen tasks.

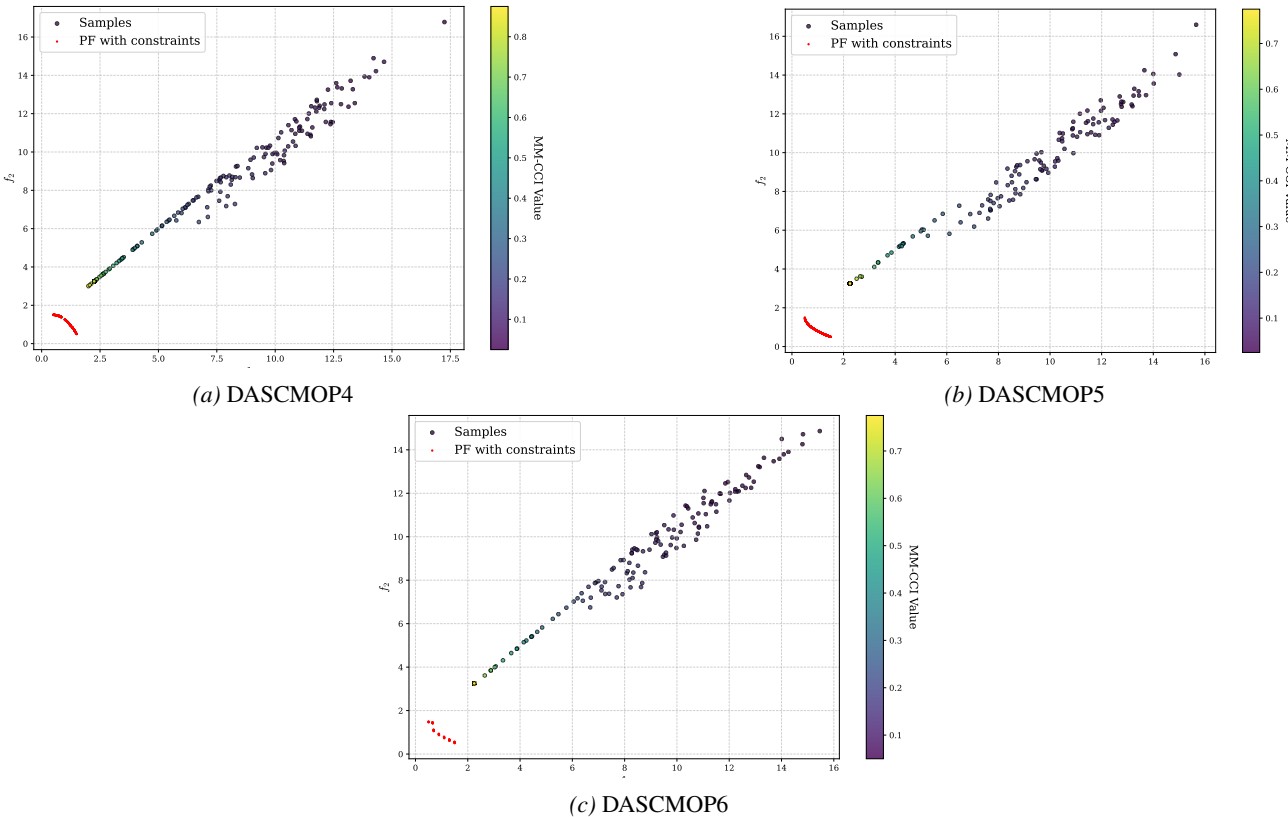

*(a)* DASCMOP4

*(b)* DASCMOP5

*(c)* DASCMOP6

*Figure 10.* Optimization performance of DAS-CMOP4, DAS-CMOP5, and DAS-CMOP6.

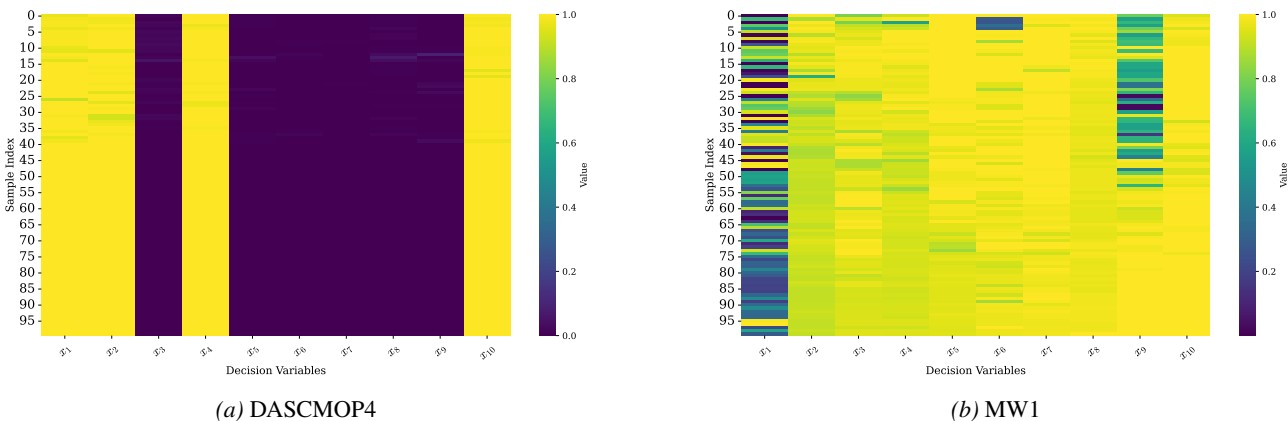

*(a)* DASCMOP4

*(b)* MW1

*Figure 11.* Comparison of Decision Variable Heatmaps during the Sampling Process between DAS-CMOP4 and MW1.

## D.4. Supplementary Analysis for RQ2

In this subsection, we provide additional analysis for RQ2, focusing on why MetaSG-SAEA may fail to find feasible solutions on DAS-CMOP4, DAS-CMOP5, and DAS-CMOP6. We observe that these instances share a key constraint function, implying highly similar constraint structures and feasible-region geometry, which likely contributes to the same failure mode across all three problems. We present the sampling process for the three problems in Figure 10, where it is observed that once a certain frontier is reached, optimization performance struggles to improve further. This indicates that although some progress is made in the early stages, the search becomes stagnated due to a lack of feasible samples to guide further exploration.

We further analyze why the search fails to break through the constraint boundary. As shown in Figure 11, we compare the decision variable heatmaps of DAS-CMOP4 and MW1 and observe that the search becomes confined to the boundary region. Since out-of-bound solutions are directly clipped, the SAEA repeatedly produces highly similar boundary-valued candidates across evaluation cycles. In contrast to MW1, this behavior effectively suppresses population diversity and reduces the search dynamics, causing the method to lose exploration vitality near the boundary.

In contrast, methods such as DRL-SAEA and EIC-MSSAEA adopt a staged optimization strategy: they first relax or ignore constraints to approach the unconstrained Pareto front, and then progressively enforce constraints to converge toward the constrained front. Such a curriculum-like process can effectively mitigate boundary-induced stagnation and improve feasibility attainment in challenging cases. An interesting direction for future work is to extend our meta-policy–based search guidance to a broader range of optimization paradigms, to further enhance generality across diverse constraint geometries.

## E. Differences from Solution Manipulation

To avoid confusion with Solution Manipulation (SM), this appendix briefly clarifies the differences between MetaSG-SAEA and SM, thereby more clearly delineating the research gap in "where to search." SM methods primarily target how to search: the meta-controller directly generates, perturbs, or refines candidate solutions in the decision space, learning solution-level update rules that are tightly coupled with (and may partially replace) the optimizer's proposal mechanism (Yang et al., 2023; Lange et al., 2024). As a result, SM operates at a solution-level granularity, specifying how concrete sampling points are produced at each step (Ma et al., 2025c; Yang et al., 2025). In contrast, MetaSG-SAEA targets where to search by performing region-level control: it learns a prioritization over MM-CCI–induced regions to allocate search effort, while leaving solution generation and the evolutionary search dynamics to the underlying SAEA. Importantly, although SM inevitably determines search locations, these locations arise implicitly from its learned solution-level proposal/update mechanism, rather than from an explicit and transferable region abstraction with a corresponding region-prioritization decision that directly optimizes the where-to-search objective, especially under tight evaluation budgets.

