# OpenReview forum: "Meta-Black-Box Optimization Can Do Search Guidance for Expensive Constrained Multi-Objective Optimization"
_ICML.cc/2026/Conference — ICML 2026 regular_

### Official Review · Reviewer_3HdG · 2026-03-07

**Soundness:** 3
**Presentation:** 3
**Significance:** 3
**Originality:** 3
**Overall Recommendation:** 5
**Confidence:** 5

**Summary:**

This paper proposes a MetaBBO framework for expensive constrained multi-objective optimization problems. A novel MM-CCI indicator is introduced to normalize constraint evaluations. Authors introduce a diffussion-based model to generate initial populations which are then optimized by a low-level SAEA optimizer. The training of the diffussion model is controlled by a DQN agent according to neural-based ELA features. Experimental results show that the proposed method surpasses baselines on some benchmarks.

**Compliance With Llm Reviewing Policy:**

Affirmed.

**Final Justification:**

The response resolves my questions, so I recommend accepting it.

**Key Questions For Authors:**

1. Can MetaSG-SAEA solve problems with equality constraints?

2. In the attention-based ELA, MM-CCI is paired with each objective value. The scale of objective values can vary across objectives and problems, which may lead to unstable training. Were these values normalized?

3. What is the motivation for designing actions to control the training of the diffusion model? How do these actions affect training and optimization, specifically, how do higher action values differ from lower ones?

4. The network structure of the DQN agent is not detailed.

5. In the zero-shot experiment, the trained MetaSG-SAEA is zero-shot transferred to the same benchmark problems used for training. I suggest it would be more insightful to train MetaSG-SAEA on MW and then zero-shot transfer it to DAS-CMOP, or vice versa. Transferring MetaSG-SAEA across problems with different numbers of objectives would also be informative.

6. Minor issue: The hyperparameters of MM-CCI and the diffusion process share the same symbols ($\alpha$ and $\beta$). It would be better to use different symbols to distinguish them.

**Limitations:**

The authors do not discuss the limitations of their work. A potential limitation could be the application scope the the proposed method since the paper does not discuss solving constrained problems with equality constraints.

**Strengths And Weaknesses:**

Strengths:
1. This paper proposes a novel inequality constraint normalization and an adaptive hyperparameter estimation method to control the hyperparameters of the normalization. Theoretical analysis for the normalization and estimation is provided.

2. An RL agent is designed to control the training of the diffusion model that generates the initial population for the low-level SAEA, enabling it to adapt to different optimization problems and optimization progress.

3. Experimental results show that the proposed method significantly outperforms the baselines. Sensitivity analysis and ablation study discuss the impact of key parameters and components in MetaSG-SAEA.

Weaknesses:
1. This paper does not discuss how to apply the proposed method to equality constraints. This may hinder the broader application of the proposed method, since a large number of constrained optimization problems in the real world involve not only inequality constraints but also equality constraints.

2. In the zero-shot experiment, the trained MetaSG-SAEA is zero-shot transferred to the same benchmark problems as those used for training, but with different dimensions. The training and testing problems are thus drawn from the same distribution. This may not adequately evaluate the cross-problem and cross-dimensional generalization ability of MetaSG-SAEA.

---

> ### Author Rebuttal · Authors · 2026-03-27
>
> We sincerely thank the reviewer for the careful reading and positive assessment of our work.  Below, we respond to the main concerns and clarify several points that may not have been sufficiently clear in the current manuscript.
>
> ## On whether MetaSG-SAEA can handle equality constraints （Q1）
> Yes. In fact, equality constraints are commonly handled in constrained optimization by converting them into inequality constraints. Specifically, an equality constraint $h(x)=0$ can be transformed into the form $$h(x)\le0, \qquad -h(x)\le0.$$ In practice, one may also introduce a small tolerance and use $$h(x)-\varepsilon \le 0,\qquad -h(x)-\varepsilon \le 0.$$ After this transformation, the resulting violations can be directly incorporated into our framework, and MM-CCI can be applied in the same way as for standard inequality constraints. Therefore, MetaSG-SAEA can be naturally extended to problems with equality constraints.
>
> ## On the zero-shot setting and cross-problem generalization (Q5)
> We apologize for the confusion.
> - Our original description may not have been sufficiently clear. Our zero-shot setting is not only cross-dimensional, but also cross-problem. Specifically, **we trained two separate meta-policies, one on the MW suite and the other on the DAS-CMOP suite, using problems with dimension $d=8$. We then directly transferred the policy trained on MW to DAS-CMOP test problems, and the policy trained on DAS-CMOP to MW test problems, both at a higher dimension $d=10$.** Therefore, our evaluation involves transfer across benchmark suites, and across dimensions, rather than only same-family cross-dimensional generalization.
> - In Line 375, we used the term “disjoint set” to briefly describe this design; we will revise the manuscript to make this setting more explicit.
>
> ## On normalization in the attention-based ELA. (Q2)
> As detailed in **Appendix C**, the objective values are normalized before being fed into the attention-based ELA, precisely to avoid instability caused by scale differences across objectives and problems. Please see Line 735 of the manuscript for the detailed description.
>
> ## On the motivation and role of the action design (Q3)
> - **Motivation.** The action is designed to let the meta-policy decide which MM-CCI region should be emphasized when training the diffusion model. In our framework, the diffusion model serves as the bridge from region-level guidance to solution-level priors.
> - **How actions affect training and optimization.**  Once an action is selected, we filter the evaluated samples by the corresponding MM-CCI interval and use only this subset to train the diffusion model. The resulting initial population is thus biased toward the selected region and passed to the lower-level SAEA. In this way, the action influences optimization indirectly but effectively by changing the region prior of the initial population, while the lower-level optimizer still preserves its own search dynamics.
> - **Difference between higher and lower action values.** Higher action values generally correspond to higher-MM-CCI regions, i.e., regions closer to feasibility, while lower action values correspond to lower-MM-CCI regions. In addition, the two action families serve different roles: $a^{(1)}$ provides broad guidance, while $a^{(2)}$ supports bounded-interval exploration, which is useful when feasible regions are disconnected or when the search risks becoming trapped locally. Please see our response to **Reviewer WV3d, Point 5,** for a more detailed explanation.
> - **Why this control is useful.** Because ECMOPs require balancing feasibility progression and multi-objective trade-offs under limited evaluations, directly deciding where the diffusion model should focus is an effective way to provide upper-level guidance without excessively restricting the lower-level optimizer.
>
> ## On Q4 and Q6.
> We sincerely thank the reviewer for the careful reading and these thoughtful comments.  We will consider further improving the manuscript in a revised version.
>
> ## Supplementary clarification on the cross-problem generalization ability
>
> - As described in **State space and attention-based ELA**, our method adopts an attention-based ELA, constructed from objective values and MM-CCI levels.
> - Owing to the attention-based design, the resulting state representation can naturally provide scalable policy inputs across tasks with varying **problem dimensions, population sizes, and numbers of objectives and constraints.**
> - Therefore, the learned meta-policy is not tied to a fixed problem instance, but is designed to generalize across heterogeneous optimization tasks.
>
> We hope that these clarifications help address the reviewer’s concerns and further support a more positive assessment of our work. We sincerely thank the reviewer again for the careful reading and positive recognition of our paper.

---

> > ### Author Rebuttal · Reviewer_3HdG · 2026-04-03
> >
> > The response addresses my concerns.

---

### Official Review · Reviewer_UeKq · 2026-03-11

**Soundness:** 3
**Presentation:** 2
**Significance:** 3
**Originality:** 3
**Overall Recommendation:** 4
**Confidence:** 3

**Summary:**

This paper proposes a Meta-Black Box optimization method for expensive constrained multi-objective optimization problems. The authors introduce MM-CCI as a metric to evaluate the degree of constraint violation of each solution in a scalar form, using a calibrated level obtained by normalizing individual constraint conditions.

In the proposed method, the meta-optimizer takes objective function values and the calibrated level as inputs, determines a search region corresponding to the calibrated level, and a diffusion model generates solutions belonging to that region. These generated solutions are used as the initial population of NSGA-II operating on a Gaussian process model. During environmental selection, NSGA-II prioritizes MM-CCI first, and then applies non-dominated sorting and crowding distance as secondary criteria.

The proposed method is evaluated on two benchmark problem suites. Models trained on one suite are tested on the other, and the results show superior performance compared to existing SAEA and meta-optimization approaches.

**Compliance With Llm Reviewing Policy:**

Affirmed.

**Final Justification:**

The proposed method introduces a novel MetaBBO approach based on constraint-level representations. Initially, there was a concern that the method might be tailored to benchmark problems with specific structural properties. However, since performance improvements were also observed on more realistic benchmark problems, I have come to regard this work as a useful and valuable contribution.

**Key Questions For Authors:**

1. Does the proposed method still function in problems where feasible and infeasible regions overlap in the objective space, as in real-world problems? If experimental validation on real-world problem suites [2–4], or a logical explanation demonstrating its effectiveness, can be provided, the evaluation may be reconsidered.

   In addition, it is necessary to demonstrate competitive performance not only against benchmark-oriented methods such as those used in the paper, but also against practically established approaches commonly used in real-world applications. For example, comparisons with surrogate-based acquisition strategies that incorporate feasibility probabilities computed from Gaussian processes (e.g., EHVI multiplied by the probability of constraint satisfaction) would be important. Showing superior or at least competitive performance against such practically adopted baselines is essential to support the claimed applicability of the proposed method beyond benchmark settings.

2. Are there prior studies on meta-optimization that focus on constraint violation measures? If so, how does the proposed method differ from them? If the originality of the proposed method can be clearly established, the evaluation may be reconsidered.

3. Why were constrained multi-objective problems specifically chosen as the target? If a clear justification is provided and it aligns with the proposed method, the evaluation may be reconsidered.

[2] Tanabe, Ryoji, and Hisao Ishibuchi. "An easy-to-use real-world multi-objective optimization problem suite." Applied Soft Computing 89 (2020): 106078.

[3] Kumar, Abhishek, et al. "A benchmark-suite of real-world constrained multi-objective optimization problems and some baseline results." Swarm and Evolutionary Computation 67 (2021): 100961.

[4] Namura, Nobuo. "Single and multi-objective optimization benchmark problems focusing on human-powered aircraft design." International Conference on Evolutionary Multi-Criterion Optimization. Singapore: Springer Nature Singapore, 2025.

**Limitations:**

It would be better to explicitly mention the possibility that the proposed method may be specialized to particular benchmark problems and may not function effectively on real-world problems.

**Strengths And Weaknesses:**

### Soundness

1. From the results in Table 1, the proposed method appears to efficiently solve the benchmark problems by controlling solution generation based on the calibrated level using MM-CCI.

### Presentation

2. The overall structure of the method can be generally understood from the paper. However, although the introduction discusses challenges in meta-optimization for expensive optimization problems in general, the actual method is specialized for constrained multi-objective problems. The paper does not clearly explain why this specific problem class was chosen.

### Significance

3. Similar to many existing studies, the proposed method appears to be specialized for benchmark problems and may have limited practical applicability. In the two benchmark suites used, feasible and infeasible regions are clearly separated in the objective space, and feasibility is effectively determined by the location in the objective space. Therefore, by using objective values and constraint violation measures as in the proposed method, feasible and Pareto-optimal solutions can be obtained efficiently.

   However, in real-world problems, feasible and infeasible solutions are often intermixed in the objective space [1]. If objective functions and constraints are uncorrelated, feasible and infeasible solutions naturally coexist in the objective space. In such cases, the proposed method may not function effectively. In this sense, the present work remains a method for efficiently solving benchmark problems rather than a broadly applicable approach.

### Originality

4. While the approach of conducting meta-optimization based on constraint violation levels using MM-CCI does not appear to be common, the reviewer has limited experience in meta-optimization and cannot definitively assess the existence of prior work. The calibrated level is essentially a normalization of constraint violation by the maximum violation, and the idea itself may be considered a natural one that many researchers could conceive.

[1] Nan, Yang, et al. "Analysis of real-world constrained multi-objective problems and performance comparison of multi-objective algorithms." Proceedings of the Genetic and Evolutionary Computation Conference. 2024.

---

> ### Author Rebuttal · Authors · 2026-03-26
>
> ## On overlapping feasible and infeasible regions in the objective space
> ### 1. Logical explanation
> We understand the reviewer’s concern as the case where there exist two decision vectors $ x_1, x_2$ such that
> $$f(x_1)=f(x_2), \quad g(x_1)\le 0, \quad g(x_2)>0,$$
> i.e., two solutions may share the same objective-space location but have different feasibility status. In such cases, feasibility cannot be inferred from objective values alone.
>  - Meta-policy directly **controls MM-CCI intervals rather than objective-space locations.** Only the evaluated samples whose MM-CCI levels fall into the selected interval are used to train the diffusion model. Therefore, even if two solutions have similar or identical objective values, they remain distinguishable through their different MM-CCI levels.
> - Our ELA is built from objective values and MM-CCI levels: the former characterize trade-offs, while the latter provides the feasibility signal.
> -  As a supplementary clarification, although the policy directly controls the MM-CCI interval, it does not stop functioning once the optimizer reaches the feasible region; please see our response to **Reviewer WV3d, Point 5,** for a more detailed explanation.
> ### 2. Experimental validation
> - We compared our method with the BO baseline EHVI×PoF on the HPA benchmark under the RQ2/RQ4 setting, selecting two CMOPs for each number of constraints (2, 3, and 4), all at difficulty level $l=1$.
> - The average log IGD ratios are 0.6034 and 0.8972 on the problems with 2 and 3 constraints, respectively, and EHVI×PoF fails to find feasible solutions on the problems with 4 constraints. This indicates that our method outperforms the BO baseline.
> - Since EHVI is more naturally aligned with the HV metric, we also evaluated HV, which led to the same conclusion.
>
> ## On originality relative to prior work
> ### 1. MetaBBO perspective
> - There is still no prior work that studies region-level search guidance in MetaBBO by explicitly focusing on constraint-violation-aware representations [1,2].
> - MetaBBO research on expensive constrained optimization remains very limited, partly because constraint-aware ELA/state representations are difficult to construct, and partly because surrogate-assisted lower-level optimizers make meta-policy training expensive due to repeated surrogate refitting [3].
> - Existing methods such as DB-SAEA [3] and DRL-SAEA [4] mainly focus on controlling **how to search**, whereas our method focuses on **where to search** through an explicit region-level meta-control mechanism.
> ### 2. Constraint-handling perspective
> - Although many existing methods use quantities such as CV, penalty functions, or feasibility ranking, they do not provide an adaptive scalar mapping for multiple heterogeneous constraints as MM-CCI does.
> - MM-CCI is not a simple penalty term or a direct replacement of CV. Instead, it adaptively **converts multiple heterogeneous constraint violations into an ordered scalar level.**
> - Together with the diffusion model, this ordered scalar level forms a region representation and guidance mechanism for the meta-policy. Its role is therefore not merely to rank feasibility, but to support region-level decisions.
> - Moreover, MM-CCI is supported by meaningful **theoretical properties, rather than being a simple normalization trick.**
>
> In short, the novelty of our work does not lie in merely “using constraint violation values,” but in transforming heterogeneous feasibility information into a structured region representation, which, together with diffusion-based initialization, enables region-level meta-control for ECMOPs.
>
> [1]"Toward automated algorithm design: A survey and practical guide to meta-black-box-optimization."TEVC (2025)
>
> [2] "Meta-Black-Box optimization for evolutionary algorithms: Review and perspective." Swarm (2025)
>
> [3] "Meta-black-box optimization with bi-space landscape analysis and dual-control mechanism for SAEA." AAAI (2026)
>
> [4] "Deep reinforcement learning assisted surrogate model management for expensive constrained multi-objective optimization." Swarm (2025)
>
> ## On focusing on constrained multi-objective problems
>
> - Our method is particularly well suited to ECMOPs, where the optimizer must use a very limited evaluation budget to handle both feasibility progression and multi-objective trade-offs.
>
> - Under this setting, deciding where to search becomes especially important, which is exactly the upper-level decision problem targeted by our framework.
>
> - In particular, MM-CCI provides a constraint-aware region signal, allowing the meta-policy to identify which regions are more worth exploring under a tight budget.
>
> - The diffusion module then translates the selected region into solution-level priors, while the lower-level SAEA preserves its own search dynamics for later Pareto refinement.
>
> In summary, ECMOPs are not an arbitrary target class, but the setting that most directly matches the **motivation** and **mechanism** of our method.

---

> > ### Author Rebuttal · Reviewer_UeKq · 2026-04-03
> >
> > Thank you for your response. Since it has become clear that the proposed method is effective even for real-world problems without specific structural assumptions, I would like to increase my score.

---

### Official Review · Reviewer_79ho · 2026-03-12

**Soundness:** 3
**Presentation:** 3
**Significance:** 3
**Originality:** 3
**Overall Recommendation:** 5
**Confidence:** 5

**Summary:**

This paper studies meta-black-box optimization for expensive constrained multi-objective optimization problems. The authors propose MetaSG-SAEA, a bi-level framework in which a meta-policy provides region-level search guidance to a surrogate-assisted evolutionary algorithm. The key idea is to learn where to search, rather than controlling how the optimizer operates. The method introduces a constraint-based region abstraction，termed MM-CCI, that maps heterogeneous constraint violations to ordered scalar levels, enabling the meta-policy to select promising regions. A diffusion model is used to generate initial populations conditioned on the selected region, and an attention-based representation of objectives and constraint levels is used as the policy state. The meta-policy is trained using reinforcement learning across multiple constrained multi-objective benchmarks. Experiments on MW and DAS-CMOP suites show improvements over several SAEA and meta-learning baselines under a limited evaluation budget.

**Compliance With Llm Reviewing Policy:**

Affirmed.

**Key Questions For Authors:**

1. How sensitive is the proposed approach to the Partition number K used in the MM-CCI representation?
2. How does the computational overhead of the framework compare with standard SAEA methods?
3. Is diffusion-based population initialization necessary, or could simpler generative mechanisms achieve similar performance?
4. How does the approach scale to higher-dimensional decision spaces or larger evaluation budgets?

**Limitations:**

The paper does not explicitly discuss the limitations, such as the computational overhead introduced by the meta-policy and diffusion model, or the scalability of the framework to higher-dimensional optimization problems.

**Strengths And Weaknesses:**

Strengths
1. The paper introduces a meaningful perspective for MetaBBO by focusing on region-level search guidance, which is particularly relevant in expensive optimization settings.
2. The proposed MM-CCI representation provides a structured way to characterize constraint violation severity and guide the search toward feasible regions.
3. The diffusion-based population initialization offers an interesting mechanism to translate region-level guidance into solution-level priors while preserving the dynamics of the evolutionary optimizer.

Weaknesses
1. The experimental evaluation is conducted on two constrained multi-objective benchmark suites (MW and DAS-CMOP). While these benchmarks are standard in constrained multi-objective optimization, additional evaluation on other benchmark families or practical expensive optimization tasks would further strengthen the empirical evidence.
2. The paper includes ablation studies on key components such as the action space design, the adaptive hyperparameter estimation in Max-CCI, and the diffusion-based population initialization. Additional analysis comparing the learned region policy with random region selection, as well as evaluating the impact of the attention-based ELA representation, could further clarify the role of these components in the framework.

---

> ### Author Rebuttal · Authors · 2026-03-27
>
> We sincerely thank the reviewer for the careful reading, constructive feedback, and positive assessment of our work.
>
> ## On the sensitivity to the Partition number K.
> As shown in Figure 7, our method is not sensitive to small variations of the Partition number K; the performance remains relatively stable across nearby K settings. This is because K is only used to provide a coarse region discretization of the MM-CCI space, rather than a finely tuned control variable. The meta-policy makes decisions at the region level, and the lower-level optimizer still preserves its own search dynamics. Therefore, moderate changes in K do not fundamentally alter the guidance behavior, which is consistent with our goal of using MM-CCI as a robust region abstraction rather than a delicate Partition scheme.
>
> ## On the computational overhead of the framework
> - We appreciate this comment. **Reviewer WV3d** also raised a similar concern, we have already started additional experiments on this aspect. At present, we have conducted a preliminary runtime analysis on several MW problems. Specifically, the ratio of diffusion runtime to the total time of one optimization cycle is 20.2%, 16.1%, and 9.3% when the number of evaluated solutions is 100, 200, and 300, respectively.
>
> - This ratio decreases as optimization proceeds because the overall cost of each cycle grows, especially due to increasing surrogate-model fitting, lower-level search, and candidate selection costs, while the diffusion overhead grows much more slowly.
>
> ## On whether diffusion-based population initialization is necessary
> - Our current ablation in Appendix D.2 already addresses this point. Replacing the diffusion module strategy (denoted as $- \mathcal{M}$ in Figure 8) leads to a clear performance drop, showing that diffusion-based warm-start is not merely an additional component, but plays an important role in translating region-level guidance into effective solution-level priors.
> - In addition, the ablations with different action variants further support that the performance gain does not come from a trivial design choice, but from the coordinated effect of the action space design and the diffusion-based initialization.
>
> ## On scalability to higher-dimensional decision spaces or larger evaluation budgets
> - **For higher-dimensional decision spaces**, our current experiments already provide initial evidence of scalability: the learned meta-policy is trained on $d=8$ problems and directly transferred to $d=10$ problems without further adaptation. Moreover, in the current literature, expensive constrained multi-objective optimization is commonly studied in relatively low-dimensional settings, and 10 dimensions is already a standard and meaningful setup in this area. This suggests that our state representation and region-level guidance mechanism are not tied to a fixed decision dimension.
>
> - **For larger evaluation budgets**, we would like to emphasize that the current work is specifically designed for evaluation-limited expensive optimization, which is also the main target setting of MetaSG-SAEA. In this setting, deciding where to search is particularly important. By contrast, if the evaluation budget is sufficiently large, the importance of deciding where to search naturally becomes less pronounced. Studying the behavior of the proposed method under larger budgets is still a meaningful direction.

---

> > ### Author Rebuttal · Reviewer_79ho · 2026-04-02
> >
> > Keep my score.

---

### Official Review · Reviewer_WV3d · 2026-03-14

**Soundness:** 3
**Presentation:** 3
**Significance:** 4
**Originality:** 3
**Overall Recommendation:** 5
**Confidence:** 3

**Summary:**

This paper proposes MetaSG-SAEA for Expensive Constrained Multi-Objective Optimization Problems (ECMOPs). The key idea is to map heterogeneous constraint evaluations into MM-CCI, a problem-agnostic scalar representation of region-level constraint information, and to build an attention-based ELA state representation from the objective values and MM-CCI levels of evaluated samples. A Double DQN-based meta-policy is then trained to select the MM-CCI interval to be prioritized by the lower-level optimizer. The evaluated samples within the selected interval are used to train a diffusion model, which generates an initial population to warm-start the lower-level SAEA. Overall, the method is designed to provide problem-agnostic search guidance for ECMOPs by learning where to search, thereby improving optimization efficiency under limited evaluation budgets.

**Compliance With Llm Reviewing Policy:**

Affirmed.

**Final Justification:**

The author's response provided sufficient clarification, so I have re-evaluated the work.

**Key Questions For Authors:**

1.    Is the attention-based ELA state representation sufficiently informative for the Double DQN-based meta-policy, given that the optimization state is highly compressed?
2.    The action space consists of only 10 discrete MM-CCI interval-based actions. Is this granularity sufficient, and does it really justify using reinforcement learning instead of a simpler policy?
3.    Could the learned region-level search guidance become counterproductive on some problem instances and actually hinder exploration?
4.    It would strengthen the paper to compare against simpler non-RL search-guidance baselines.
5.    While MM-CCI seems useful for moving the search from infeasible to feasible regions, might it become less helpful-or even restrictive-once the search is already in the feasible region and needs finer Pareto refinement?
6.    The diffusion-model-based warm-start may introduce substantial computational overhead, but the paper does not clearly discuss whether this additional cost is justified.
7.    From an efficiency perspective, it would be useful to compare the diffusion module with simpler and cheaper alternatives

**Limitations:**

1.    The paper does not theoretically justify whether the proposed reward function-which combines MM-CCI progress and IGD improvement-is well aligned with improvement toward the feasible Pareto front.
2.    While the 10-action MM-CCI interval-based action space makes learning easier, it may be too coarse to capture nuanced region-selection behavior, and the paper does not fully justify this design choice.
3.    The empirical study is limited to the MW and DAS-CMOP benchmark suites under relatively restricted dimensional settings, so the broader generalization of MetaSG-SAEA remains to be further validated.

**Strengths And Weaknesses:**

(Strengths)

(1) The paper introduces a meaningful shift from learning how to search to learning where to search through region-level search guidance.

(2) MM-CCI is not merely heuristic but is supported by a reasonably clear theoretical motivation and stated mathematical properties.

(3) The method shows promising zero-shot transfer performance across both problems and dimensions.

(4) The empirical section is thorough, covering convergence, generalization, consistency, sensitivity, ablations, and model complexity.

(Weaknesses)

(1) The method fails to identify feasible solutions on DAS-CMOP4/5/6, suggesting a limitation on some hard constrained landscapes.

(2) The design of the state representation and discrete action space appears somewhat heuristic, and the paper could better justify these design choices.

---

> ### Author Rebuttal · Authors · 2026-03-25
>
> ## On whether the attention-based ELA state is overly compressed
> - Compact ELA states are common in MetaBBO, typically with only 4–20 dimensions, as in DRL-SAEA and DB-SAEA; our design is therefore not unusually compressed.
> -  Meta-policy only performs region-level guidance with an action space of just 10 discrete actions, rather than complex control over the lower-level optimizer, so a high-dimensional state is unnecessary.
> - As stated in Line 326, the **ELA is built from objective values and MM-CCI levels**: the former provide multi-objective trade-off information, while the latter provides the feasibility signal. These are exactly the key signals needed for search guidance. Since they do not involve decision-space landscape information, the resulting representation is not inherently complex and does not require high dimensionality.
>
> ## On the 10-action discrete MM-CCI interval-based action space
> -  Since ECMOPs are evaluation-limited, our meta-policy essentially controls **the selection of a subset from a small sample set for diffusion-model training.** In this setting, a finer action granularity is unlikely to bring meaningful benefit; for example, when selecting from a set of only 10 samples, choosing the top 10% versus 13% leads to the same subset.
> - As illustrated in Fig 3, $a^{(1)}$ provides broad guidance, while $a^{(2)}$ is designed to help the optimizer avoid becoming trapped in local optima. Please see our **response to Point 5** for a more detailed explanation. In contrast, a simpler policy such as a hand-crafted rule is unlikely to capture the distinct roles of these two action types, and usually transfers poorly across problems.
> - A relatively coarse action space is also better suited for region-level guidance, as it is more **robust and less likely to overcommit to a mistakenly preferred small region.**
>
> ## On whether the learned region-level search guidance could hinder exploration
> - The region-level guidance is designed as a **soft and robust bias rather than a hard restriction on the search process;** the lower-level SAEA still preserves its own evolutionary search dynamics and exploration ability.
> - The meta-policy also supports exploratory guidance. For example, $a^{(2)}$ selects a **bounded MM-CCI interval rather than always favoring higher-MM-CCI samples.**
>
> ## On the suggestion to compare with simpler  non-RL search-guidance baselines
> We thank the reviewer for this helpful suggestion. We conducted an additional ablation using a simple non-RL rule that uniformly shifts the selected MM-CCI interval toward 1 as the number of evaluations increases. The result shows a clear performance drop, with the average log IGD ratio reaching 0.4713.
>
> ## On whether MM-CCI may become less helpful after reaching the feasible region
> - It remains useful after reaching the feasible region. Many constrained multi-objective problems have disconnected feasible regions (e.g. Figure 3a), so methods may find one feasible region and then become trapped locally. Our ELA is built from objective values and MM-CCI levels jointly: the objective values provide trade-off information, while MM-CCI provides the feasibility signal. Therefore, **even after reaching one feasible region, the policy can still use actions such as $a^{(2)}$ to guide the optimizer toward other directions and other feasible regions (e.g. Figure 3c).**
> - It is not restrictive. As emphasized in Line 74, our goal is to **provide effective guidance without excessive restriction.** Our method is only used to prioritize warm-start regions, while the lower-level SAEA still preserves its own search dynamics for later Pareto refinement.
>
> ## On the computational overhead of the diffusion-model-based warm-start
> - We appreciate this comment. We have already started additional experiments on this aspect. At present, we have conducted a preliminary runtime analysis on several MW problems. Specifically, the **ratio of diffusion runtime to the total time of one optimization cycle** is 20.2%, 16.1%, and 9.3% when the number of evaluated solutions is 100, 200, and 300, respectively.
> -  This ratio decreases as optimization proceeds because the overall cost of each cycle grows, especially due to increasing surrogate-model fitting, lower-level search, and candidate selection costs, while the diffusion overhead grows much more slowly.
>
> ## On comparing the diffusion module with simpler and cheaper alternatives.
> Our ablation in **Appendix D.2** already partially addresses this point: replacing the diffusion module (denoted as $−\mathcal{M}$ in Fig 8) leads to a substantial performance drop.
>
> We hope that these clarifications help resolve the reviewer’s concerns and provide a clearer picture of the contribution of our work. We sincerely thank the reviewer for the careful reading and positive recognition of our paper.

---

> > ### Author Rebuttal · Reviewer_WV3d · 2026-04-03
> >
> > The author's answer has resolved my questions.

---

### Decision · Program_Chairs · 2026-04-30

**Decision:**

Accept (regular)

**Comment:**

This paper proposes a meta black-box optimization method for expensive, constrained multi-objective optimization problems. The authors introduce max–min constraint-calibrated inequality (MM-CCI) as a metric of region-level constraint information. The key point of this paper is to learn where to search through region-level search guidance, rather than learning how to search. The experimental evaluation using benchmark problem suites demonstrates that the proposed method outperforms existing surrogate-assisted evolutionary algorithms (SAEA) and meta-optimization approaches.

The motivation, shifting from learning how to search to learning where to search, is meaningful, and the introduced technique, MM-CCI, is technically sound. As a limitation, the proposed method fails to identify feasible solutions in several problems. As the experimental evaluation is limited to two benchmark problem suites, evaluating the proposed method on real-world problem suites will strengthen the paper.

All reviewers recognize the significance and the effectiveness of the paper and give positive scores. Therefore, I would recommend accepting this paper.

As the author-reviewer discussion on the rebuttal phase is fruitful, I would recommend that the authors include additional explanations based on the reviewers' comments.　In addition, as an AC comment, I highly recommend that the authors release the code for the experiments, as the proposed method seems relatively complicated, which would be beneficial for the research community to reproduce the results and further develop this line of research.